# SciFIBench: Benchmarking Large Multimodal Models for Scientific Figure Interpretation

**Jonathan Roberts**
University of Cambridge
jdr53@cam.ac.uk

**Kai Han**
The University of Hong Kong
kaihanx@hku.hk

**Neil Houlsby**
Google DeepMind
neilhoulsby@google.com

**Samuel Albanie**
samuel.albanie.academic@gmail.com

## Abstract

Large multimodal models (LMMs) have proven flexible and generalisable across many tasks and fields. Although they have strong potential to aid scientific research, their capabilities in this domain are not well characterised. A key aspect of scientific research is the ability to understand and interpret figures, which serve as a rich, compressed source of complex information. In this work, we present **SciFIBench**, a scientific figure interpretation benchmark consisting of 2000 questions split between two tasks across 8 categories. The questions are curated from arXiv paper figures and captions, using adversarial filtering to find hard negatives and human verification for quality control. We evaluate 28 LMMs on SciFIBench, finding it to be a challenging benchmark. Finally, we investigate the alignment and reasoning faithfulness of the LMMs on augmented question sets from our benchmark. We release SciFIBench to encourage progress in this domain: https://SciFIBench.github.io/.

## 1  Introduction

Lately, the rate of progress in the development of artificial intelligence (AI) has significantly increased. The emergence of foundation models [1], trained on large-scale broad data using extensive computational resources enabling generalisation across many downstream applications, has greatly expanded the range of possible domains and tasks in which machine intelligence can operate. Notable large language models (LLMs), such as GPT-4 [2], LLaMA [3], and PaLM [4], and subsequent large multimodal models (LMMs), for example, GPT-4V [5], Qwen [6], and Gemini [7], have proven to be flexible and generalisable across many tasks. In particular, their capabilities have been demonstrated in fields such as mathematics [8, 9, 10], medicine [11, 12, 13, 14, 10], and finance [15, 16], as well as writing code [17] and the geographic and geospatial domains [18, 19].

One area that is beginning to receive more attention is the *scientific domain*, which has the potential to greatly benefit from AI tooling. Although the current generation of frontier models is arguably unable to perform independent, end-to-end scientific research, there is an emerging body of evidence [17, 10, 20, 21, 22, 23] suggesting they can be used as a tool to assist different stages of the scientific process. A key aspect of scientific research is the ability to *understand figures*, which serve as a rich, compressed source of complex information. As noted in [24], unique challenges arise from the complex and dense semantics of scientific images and the sophisticated language preferences of researchers. While the abilities of LMMs across some domains are relatively well-understood thanks to established benchmarks [25, 26, 27, 28, 29], their capacity to understand scientific figures is not well known. However, reliably characterising the ability of a model to interpret scientific figures is challenging without an obvious objective evaluation metric. Another consideration is the source of accurate ground truth; manually annotating a sufficiently large evaluation set of figures with accurate descriptions is unfeasible, and challenging without appropriate domain knowledge.

38th Conference on Neural Information Processing Systems (NeurIPS 2024) Track on Datasets and Benchmarks.

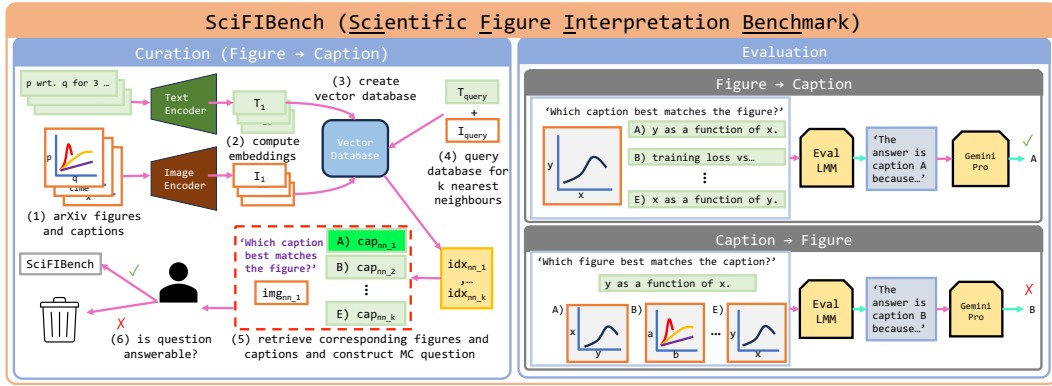

Figure 1: **Overview of SciFIBench.** *Left*: our benchmark consists of 2000 multiple-choice scientific figure interpretation questions curated from arXiv papers using adversarial filtering and human verification to maximise difficulty and quality, respectively. *Right*: we evaluate a suite of LMMs on the two core SciFIBench tasks, leveraging an LLM for automatic evaluation.

We circumvent these issues by reframing the evaluation to a multiple-choice setting, using the figure captions as ground truth descriptions – see Fig. 1. Concretely, using figure-caption pairs from arXiv papers, we construct a pool of multiple-choice questions for the two tasks shown in Fig. 1. Following other popular works [30], we adopt adversarial filtering when curating the negatives for each question to increase the difficulty. To further improve the quality, we utilise human verification on *every* question to ensure they are maximally answerable. We create **SciFIBench** (Scientific Figure Interpretation Benchmark) by sampling from this question pool with the following three objectives in mind: (1) **Quality** – we perform human verification on every question to ensure high-quality questions that are answerable. (2) **Efficiency** – we choose a small-scale set of questions, enabling streamlined evaluation and ensuring the benchmark can maximally be used by the community. (3) **Robustness** – we conduct careful analysis to verify SciFIBench offers a suitably robust evaluation. Our benchmark consists of 2000 high-quality, challenging questions.

We evaluate a suite of 28 open- and closed-source LMM baselines on SciFIBench and compare the performance to human and vision-language model (VLM) baselines. To overcome the challenges associated with post-processing the output of LMMs to extract a specific answer at scale, we leverage Gemini-Pro [7] to parse the output of all evaluated LMMs and extract the relevant multiple-choice letter answers, enabling automatic evaluation. Finally, we carry out preliminary experiments probing the alignment and faithfulness of the LMMs when answering questions in our benchmark. We hope our insights will encourage further research in this direction.

To conclude, our main contributions are as follows: (i) We curate **SciFIBench** to evaluate scientific figure interpretation. (ii) We benchmark 28 LMMs on SciFIBench and compare the performance to human and VLM baselines. (iii) We introduce an experimental setting probing the instruction-following abilities and faithfulness of reasoning of the LMMs. (iv) We release SciFIBench to drive progress in LMM scientific figure interpretation and understanding research. We derive these key insights from our work:

- SciFIBench proves to be a challenging benchmark for current LMMs.
- GPT-4o [31] and Gemini 1.5 [32] are the best-performing models, outperforming all the VLM baselines but are beaten by the human baseline.
- Adversarial filtering significantly increases multiple-choice question difficulty but human filtering is crucial to ensure high-quality, answerable questions.
- Leveraging a strong LLM to evaluate the noisy output of the evaluated LMMs proves accurate and viable for automatic evaluation.
- The evaluated LMMs show varying levels of faithfulness in their answers.

## 2   Related Work

**Scientific Figure Interpretation.** Several approaches have been proposed to investigate the capacity of multimodal models to interpret scientific figures. These include **question answering** benchmarks such as ChartQA [33], PlotQA [34], and FigureQA [35], which ask complex reasoning questions about scientific figures. ACL-Fig [36] introduces the task of **type classification** for scientific figures. A

large body of literature exists that evaluates the quality of **generated captions** for scientific figures. The progenitor for many subsequent works is SciCap [37], in which an image-captioning model is trained to generate high-quality captions. SciCap+ [38] builds this idea further and includes figure mention-paragraphs in addition to input figures. SciCap-Eval [39] investigates the usage of LLMs for ranking scientific figure captions. VisText [40] fine-tunes language models to generate captions for scientific charts, and FigCaps-HF [41] introduces a framework that initially learns a human feedback prediction model and incorporates this to optimise caption generation based on reader preference. The SciMMIR benchmark [24] characterises the abilities of vision-language models to understand scientific figures through **retrieval** experiments. More recently, a few works [5, 10] have conducted a qualitative analysis of LMM (specifically, GPT-4V [5]) performance on a small handful of scientific figures. We draw inspiration from these works, incorporating some of the methodological ideas. However, our work focuses on a quantitative evaluation of LMMs for the task of understanding scientific figures, which has yet to be reported. We also re-frame the task to a multiple-choice setting as this is more suitable for robust evaluation of LMMs.

**LMM Benchmarks.** A number of benchmarks aimed at multimodal model evaluation have been developed in recent years. Prominent natural image benchmarks include LVLM-eHub [42], MMBench [26], MME [43], MM-Vet [44], and SEEDBench [28] and SEEDBench-2 [29], which both consist of multiple-choice questions across different domains and evaluation dimensions. A small-scale geographic and geospatial benchmark is introduced in [19]. LAMM [45] evaluates a variety of computer vision tasks on 2D natural images as well as 3D point clouds. Other benchmarks, such as HallusionBench [46], focus on the failure modes and hallucinations of the models. MathVista [27] introduces a mathematical reasoning in visual contexts metadataset, which includes scientific figures and charts. This benchmark contains similar image types to our work but has a different focus and uses different question types. The MMMU benchmark [25] includes multi-discipline college-level image-based problems and questions. Although limited to text, we take inspiration by the adversarial filtering approach taken in [30], in the curation of the multiple-choice questions in our work. Our work incorporates stylistic and methodological inspiration from these works but tackles a different image type with a different overall focus of scientific figure interpretation.

## 3 SciFIBench

SciFIBench is comprised of 2000 questions, derived from figures and captions extracted from arXiv papers, curated into two tasks and split into 2 subsets based on the data source (Tab. 1).

| Subset | # questions per task | | Num. pairs in pool | Num. categories | Source data | Human verified? |
|---|---|---|---|---|---|---|
| | Figure→Caption | Caption→Figure | | | | |
| **CS** | 500 | 500 | 94k | 1 | [37] | ✓ |
| **General** | 500 | 500 | 102k | 7 | [47] | ✓ |

Table 1: **SciFIBench** is composed of two subsets of questions based on category and source.

### 3.1 Tasks

SciFIBench consists of the following two tasks related to scientific figure interpretation (Fig. 1):
> **Figure→Caption**: Given an input figure, along with a set of 5 captions labeled A-E, select the correct caption for the figure.
> **Caption→Figure**: Given an input caption, along with a set of 5 figures labeled A-E, select the correct figure for the caption.

### 3.2 Curation methodology

We use the SciCap dataset [37] as our initial source of scientific figure-caption pairs. SciCap is a large-scale dataset consisting of figures and corresponding captions extracted from arXiv computer science (CS) papers between the years 2010-2020. From SciCap, we select the *Single-Sentence* subset (train, val, test), containing ∼94k figure-caption pairs, and only includes captions that are one sentence in length. The figures are filtered to remove any containing subfigures, and the captions are normalised to remove figure numbers. We then perform the following preprocessing and curation steps:
1. **Deduplication**: We initially drop any captions (and corresponding figures) if they are duplicates.

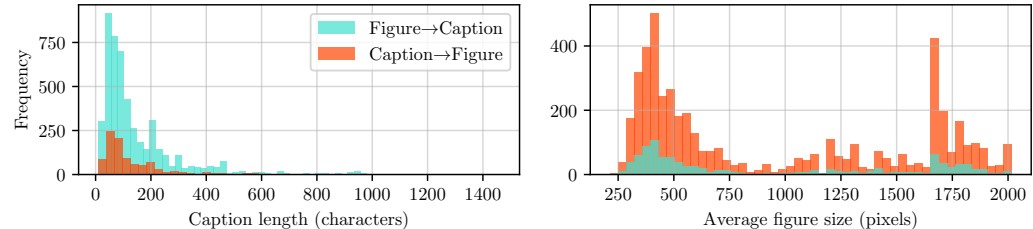

Figure 2: **SciFIBench question figure and caption statistics**.

2. **Compute embeddings**: We then use a variant of CLIP [48][1]. to compute embeddings for each figure-caption pair. After normalising, we concatenate the text and image embeddings to form joint embeddings, represented as vectors $x \in \mathbb{R}^d$, where $d$ is equal to 2048.

3. **Construct vector database**: Using Faiss [51], we create a vector database of the joint embeddings.

4. **Find nearest neighbours**: For each embedding, we search for the $k$ nearest neighbours based on Euclidean distance. Concretely, given the set of database embeddings $\{x_i, i=1..N\} \subset \mathbb{R}^d$ and a query embedding $q \in \mathbb{R}^d$, we compute the $k$ nearest neighbours of $q$ as:

$$(n_1,...,n_k) = \operatorname*{argmin}_{n=1..N}^{k} ||q - x_n||. \tag{1}$$

5. **Similarity filtering**: To increase the likelihood the multiple-choice questions are answerable we remove very similar figure-caption pairs from our dataset (*e.g.*, with minor formatting differences but no semantic difference) by dropping a sample $(x_s)$ if its distance to the query embedding (*i.e.*, $||q - x_s||$) falls below a threshold.

6. **Question construction**: For each selected figure-caption pair, we create multiple-choice questions using the $k$ nearest neighbours. For the **Figure→Caption** task, we create target captions by randomly shuffling the true caption with the corresponding $k$ nearest neighbour captions. Similarly, for the **Caption→Figure** task, we create the target figures by randomly shuffling the true figure with the corresponding $k$ nearest neighbour figures.

7. **Categorisation**: We categorise questions based on the arXiv category of the true figure-caption pair. Questions in the 10 most common categories are grouped individually while those in less common categories are labelled 'other cs'; questions from cross-listed papers are labelled 'cross-list'.

8. **Difficulty filtering**: We adopt the average distance of the joint embeddings of the negatives to the true answer as a measure of question difficulty. We sort the questions based on this difficulty.

9. **Human verification**: We sample the most difficult questions per category and perform human verification to select 'answerable' questions. We classify a question as answerable if it contains sufficient information for a domain expert to determine the single correct answer (*i.e.*, questions with ambiguous choices; or references to context-dependent details, such as 'Exp. 1', 'Config. 1', etc. are disregarded). Minor text edits were made for a small subset of the questions to reduce ambiguity.

Following these steps, we obtain a pool of high-quality questions. We evaluate GPT-4V [5] and Gemini-Pro Vision [7] on the pool and select questions that either model answers incorrectly and sample the remaining questions per category to create our curated 'CS' question set. As some categories had few answerable questions in the pool, category balance was approached, but not achieved in all cases – Fig. 3 illustrates the category representation. For example, the pool of possible 'cs.AI' category questions was dominated by figures/captions from a single paper; to avoid introducing bias, we only included 10 such questions per task. For analysis, a noisier subset was then constructed by taking the next 5000 most difficult questions per task, sampled across categories, without human checking. Example questions from the curated set are shown in Fig. 5. To expand the diversity of the questions in our benchmark, we utilised the ArXivCap dataset [47], which contains figure-caption pairs from papers covering 32 arXiv domains up to 2023. Due to its larger scale, we initially randomly downselected 25% of the data and removed all pairs from CS papers. We then repeated the curation steps outlined above, to create a pool of answerable questions for each task. We then carried out a category-balanced downsampling of questions Gemini 1.5 Pro [32] and GPT-4o [31] answered incorrectly to reach a final curated 'General' set of 500 questions per task. In Fig.2 we show figure and caption statistics for questions across both subsets of SciFIBench. **Quality control.** We focus our evaluation on a small set of questions to ensure high quality. Having

---

[1]We use ViT-H-14-378-quickgelu [49] as it attains strong zero-shot performance on numerous datasets [50].

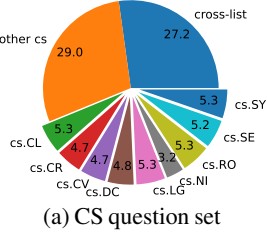

(a) CS question set

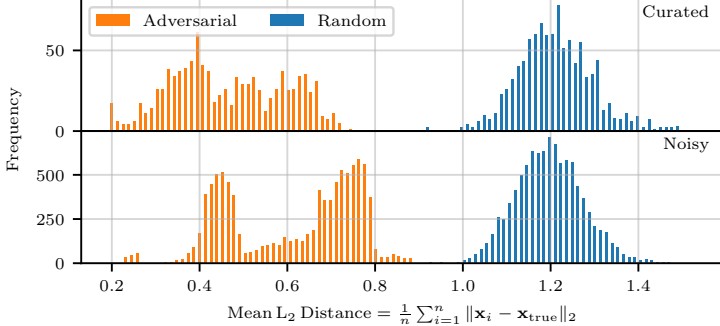

Figure 4: **Difficulty distribution comparison of the *Noisy* uncurated questions (5000) and the *Curated* questions (1000) included in the CS set of SciFIBench**. We gauge difficulty using the mean $L_2$ distance between the joint embeddings ($\mathbf{x}$) of the positive and negative answers for each question. A higher distance indicates an easier question.

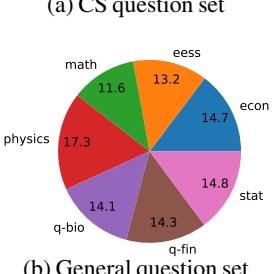

(b) General question set

Figure 3: **SciFIBench category representation.**

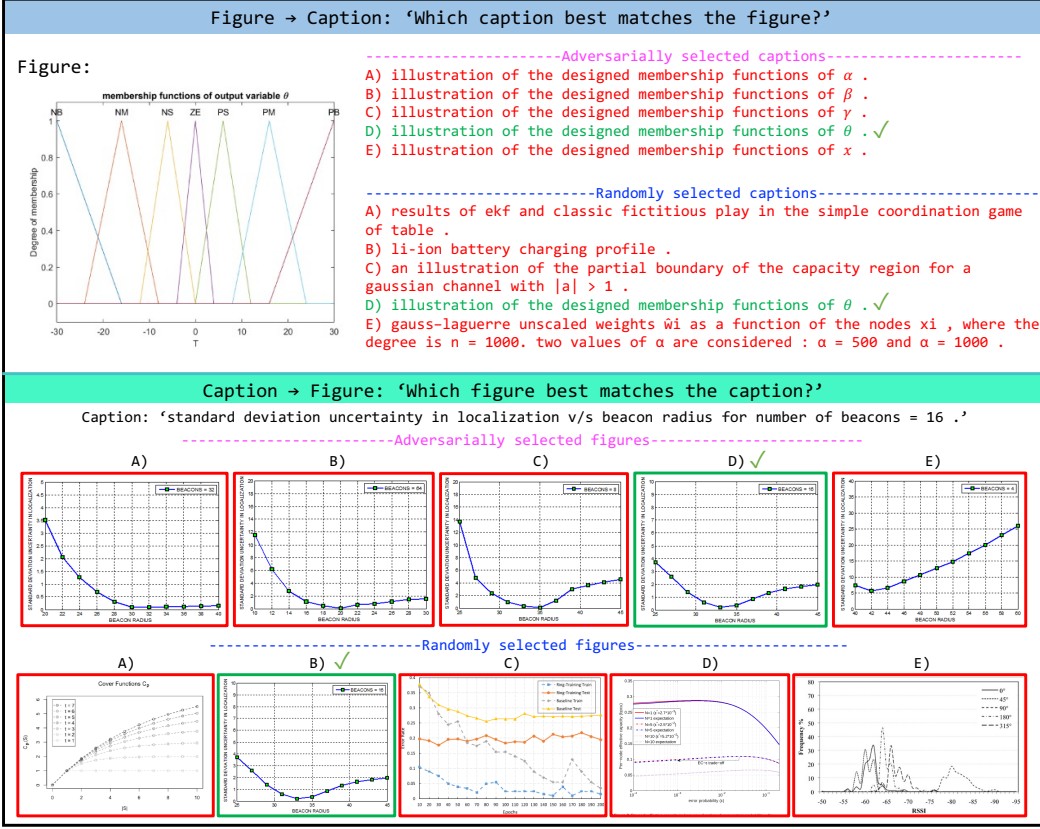

Figure 5: **Example SciFIBench questions** for each task with the challenging adversarially-selected and easier randomly-selected negatives. SciFIBench covers a broad range of figure types including line/pie/bar/flow charts, scatter/box/3D/contour plots, multiplots, maps, heatmaps, and decision trees.

manually checked each question, we conservatively estimate the noise level in the curated set – *i.e.* included in SciFIBench – to be at most a few percent. In these minority, questionable cases, we estimate there is a reasonable chance the questions can be answered with appropriate domain expertise. Based on spot checks, we estimate the noise level on the noisy, uncurated questions to be ∼20-25%. Minor cosmetic errors, such as typos in captions or obscured axis labels, originating from the original data were

deliberately left unchanged when included in SciFIBench to increase realism and difficulty. **Question difficulty.** Preliminary ablation studies on a random set of questions showed that, for nearly all the LMMs evaluated, selecting hard negatives using nearest neighbours determined by joint-embedding similarity yields the most challenging questions, with lower accuracy scores than the single-modality neighbours. Fig. 4 outlines a comparison of the difficulty distribution of the included curated questions and uncurated noisy questions, based on $L_2$ distance. The effect of adversarial compared to random negatives selection can be seen in the disparity of the orange and blue distributions, with the adversarial negatives having a much lower mean $L_2$ distance and therefore higher difficulty. As expected from the curation process, the curated adversarial distribution is more challenging than the noisy distribution.

## 4 Experiments

Through a variety of experiments, we evaluate the scientific figure interpretation abilities of a selection of LMMs on our SciFIBench benchmark and conduct a detailed analysis of their performance.

### 4.1 Baselines

**LMMs.** We evaluate the following **closed-source** models: GPT-4 {V, Turbo, o} [5, 52, 31] Gemini-Pro Vision [7], Gemini 1.5 {Flash, Pro} [32], and Claude 3 {Opus, Sonnet and Haiku} [53]. We also evaluate the following **open-source** models: IDEFICS [54], Qwen-VL [6], Emu2 [55], TransCore-M [56], InternLM-Composer 1,2 [57], CogVLM [58], OmniLMM [59], Yi [60], InstructBLIP [61], Monkey [62], and LLaVA-1.5 [63]. We use chat / instruction-tuned variants of each model (rather than base models) and compare the performance of multiple model sizes where available. Roughly half of these baselines can take interleaved text and images as input, and therefore be evaluated on the Caption→Figure task. We also consider a text-only baseline in which we provide the LMM with the output from an OCR model [64] rather than images. **VLMs.** As a point of comparison, we evaluate strong VLM models on SciFIBench. Specifically, we evaluate a MetaCLIP [65] variant, the Google Multimodal Embedding Model [66], and the CLIP model [49] used to determine the nearest neighbour multiple-choice options. **Humans.** Additionally, we evaluate a human baseline to gauge the relative performance difference between humans and LMMs. The humans (undergraduate and postgraduate students) were presented with the same prompt as the models.

While it is difficult to say with certainty if arXiv data was included in the training sets of these models, there might be some leakage, as expected when using web images. However, given the scale of the training data, we do not expect this to impact our evaluation.

### 4.2 Experimental Settings

**Inference.** For the closed-source models, inference was carried out via the OpenAI API [67] or Vertex AI API [68]. We use Transformers [69] and OpenCompass toolkit [70] to access the open-source models and conduct inference using NVIDIA A100 GPUs. With current pricing, evaluating GPT-4V on SciFIBench costs ∼$30. For the open-source models, the typical inference runtime using an A100 is ∼1 hour (*e.g.*, using Qwen-VL).

**Hyperparameters.** We select model hyperparameters that produce deterministic output. For the open-source models, we utilise the greedy search decoding strategy, in which the most probable token is selected from the model vocabulary $V$ at each step, conditional on the preceding tokens *i.e.*, $w_{n+1} = \arg\max_{w \in V} P(w|w_1, w_2, ..., w_n)$. For the Gemini and Claude models, we set the $temperature$ to 0 and $topk$ to 1; for the GPT-4 models, we also set the $temperature$ to 0.

**Prompting.** We adopt a generic 0-shot chain-of-thought [71] style prompt for each task, details of which can be found in the Appendix. Where relevant, we follow model-specific prompting suggestions and modify the prompt template accordingly. We found that shuffling the order of the multiple-choice answers causes performance to vary within a range of 5%.

**Automatic Evaluation.** Despite instruction to constrain the format of the model answers to each question to just the target choice letter, *e.g.*, 'A', most of the evaluated models did not consistently follow this, posing a challenge to automatic evaluation. To overcome this, we used Gemini-Pro to initially parse the output and extract the answer letter or flag if no single answer was given.

| Model | CS Fig.→Cap. Advers. negatives | | Random negatives | CS Cap.→Fig. Advers. negatives | | Random negatives | General Fig.→Cap. Advers. negatives | General Cap.→Fig. Advers. negatives | Overall Fig.→Cap. Advers. negatives | Overall Cap.→Fig. Advers. negatives |
|---|---|---|---|---|---|---|---|---|---|---|
| *Closed-source LMMs* | | | | | | | | | | |
| GPT-4V [5] | 69.4 | +29.8 | 99.2 | 58.4 | +38.0 | 96.4 | - | - | - | - |
| GPT-4 Turbo [52] | 68.0 | +30.6 | 98.6 | 60.6 | +36.8 | 97.4 | 62.8 | 55.2 | 65.4 | 57.9 |
| GPT-4o [31] | **75.4** | +24.2 | **99.6** | 72.2 | +26.8 | **99.0** | **72.2** | 58.6 | **73.8** | 65.4 |
| Gemini Pro Vision [7] | 56.0 | +41.2 | 97.2 | 52.4 | +46.0 | 98.4 | 50.6 | 39.6 | 53.3 | 46.0 |
| Gemini 1.5 Pro [32] | 74.0 | +25.0 | 99.0 | **76.0** | +22.4 | 98.4 | 65.2 | 56.2 | 69.6 | **66.1** |
| Gemini 1.5 Flash [32] | 74.4 | +24.6 | 99.0 | 69.6 | +29.4 | **99.0** | 65.8 | **62.4** | 70.1 | **66.1** |
| Claude 3 Haiku [53] | 52.6 | +36.4 | 89.0 | 43.8 | +34.6 | 78.4 | 52.6 | 33.0 | 52.6 | 38.4 |
| Claude 3 Sonnet [53] | 53.4 | +33.0 | 86.4 | 58.4 | +31.6 | 90.0 | 53.6 | 55.0 | 53.5 | 56.7 |
| Claude 3 Opus [53] | 59.8 | +27.0 | 88.2 | 49.2 | +32.0 | 81.2 | 50.8 | 47.4 | 55.3 | 48.3 |
| *Open-source LMMs* | | | | | | | | | | |
| IDEFICS-9b-Instruct [54] | 20.6 | +4.4 | 25.0 | 20.2 | -3.0 | 17.2 | 17.6 | 12.6 | 19.1 | 16.4 |
| IDEFICS-80b-Instruct [54] | 20.6 | +17.6 | 38.2 | **24.2** | +0.4 | 24.6 | 18.4 | **20.6** | 19.5 | **22.4** |
| Qwen-VL-Chat [6] | 28.0 | +30.0 | 58.0 | 16.0 | +1.0 | 17.0 | 17.0 | 19.2 | 22.5 | 17.6 |
| Emu2 [55] | 20.8 | +28.4 | 49.2 | - | - | - | 19.6 | - | 20.2 | - |
| TransCore-M [56] | **51.0** | +28.2 | **79.2** | - | - | - | **27.4** | - | **39.2** | - |
| InternLM-XComposer-7b [57] | 34.0 | +21.6 | 55.6 | - | - | - | 21.6 | - | 27.8 | - |
| InternLM-XComposer2-7b [57] | 28.0 | +46.0 | 74.0 | - | - | - | 23.8 | - | 25.9 | - |
| CogVLM-Chat [58] | 40.8 | +17.0 | 57.8 | - | - | - | 24.0 | - | 32.4 | - |
| OmniLMM-3b [59] | 35.8 | +29.0 | 64.8 | - | - | - | 24.8 | - | 30.3 | - |
| OmniLMM-12b [59] | 34.2 | +34.0 | 68.2 | - | - | - | 27.2 | - | 30.7 | - |
| Yi-VL-6b [60] | 41.4 | +30.4 | 71.8 | - | - | - | 27.0 | - | 34.2 | - |
| Yi-VL-34b [60] | 32.6 | +29.4 | 62.0 | - | - | - | 21.4 | - | 27.0 | - |
| InstructBLIP-FlanT5-xl [61] | 35.8 | +22.2 | 58.0 | - | - | - | 19.0 | - | 27.4 | - |
| InstructBLIP-FlanT5-xxl [61] | 36.2 | +20.4 | 56.6 | - | - | - | 26.8 | - | 31.5 | - |
| InstructBLIP-Vicuna-7b [61] | 21.0 | -3.4 | 17.6 | - | - | - | 12.8 | - | 16.9 | - |
| InstructBLIP-Vicuna-13b [61] | 22.2 | +5.2 | 27.4 | - | - | - | 15.6 | - | 18.9 | - |
| Monkey-Chat [62] | 27.2 | +22.8 | 50.0 | - | - | - | 18.2 | - | 22.7 | - |
| LLaVA-1.5-7b [63] | 32.8 | +27.8 | 60.6 | - | - | - | 22.8 | - | 27.8 | - |
| LLaVA-1.5-13b [63] | 25.0 | +41.2 | 66.2 | - | - | - | 20.2 | - | 22.6 | - |
| *Text-only input* | | | | | | | | | | |
| Gemini-Pro 1.5 Flash [32] | 48.0 | +33.4 | 81.4 | 39.2 | +34.4 | 73.6 | 51.0 | 35.8 | 49.5 | 37.5 |
| *VLMs* | | | | | | | | | | |
| CLIP ViT-H-14-378-quickgelu [50] | 41.8 | +50.6 | 92.4 | 42.6 | +53.4 | 96.0 | **30.6** | **30.0** | 36.2 | 36.3 |
| MetaCLIP ViT-H-14-quickgelu [65] | 36.6 | +53.2 | 89.8 | 35.4 | +54.8 | 90.2 | 24.2 | 25.2 | 30.4 | 30.3 |
| Google Multimodal Embedding [66] | **47.6** | +46.2 | **93.8** | **54.4** | +44.0 | 98.4 | 28.2 | 28.4 | **37.9** | **41.4** |
| *Human (25 questions per task*)* | | | | | | | | | | |
| Human ($\mu \pm \sigma$) | **86.4**±8.24 | +10.7 | **100.0** | **78.4**±8.24 | +22.7 | **100.0** | - | - | - | - |
| GPT-4o | 72.0 | +28.0 | **100.0** | 76.0 | +24.0 | **100.0** | - | - | - | - |
| Gemini-Pro 1.5 | 84.0 | +16.0 | **100.0** | 72.0 | +28.0 | **100.0** | - | - | - | - |
| CLIP ViT-H-14-378-quickgelu | 48.0 | +44.0 | 92.0 | 56.0 | +44.0 | **100.0** | - | - | - | - |
| TransCore-M | 36.0 | +48.0 | 84.0 | - | - | - | - | - | - | - |

Table 2: **Performance on SciFIBench**. Results for questions with adversarial and randomly-selected negatives are shown for the CS question set, along with the difference between them. *25 questions per task were randomly selected for the human baseline experiments with model scores shown for the same subset of questions. For the adversarial negatives, the human score is calculated as a mean of 5 participants, while only one human conducted the random negatives evaluation.

## 4.3 Main Results

To gauge the abilities of frontier LMMs to interpret scientific figures, we evaluate a diverse set of LMMs and other baselines on SciFIBench, the results for which are displayed in Tab. 2. Note, our core analysis is in reference to results obtained on the adversarially generated question negatives. We present our key findings as follows:

**SciFIBench represents a difficult benchmark.** The best-performing models, GPT-4o and Gemini 1.5 Flash, attain scores of 73.8% and 70.1% for the Figure→Caption task, and 65.4% and 66.1% for the Caption→Figure tasks, respectively. This shows that even at the frontier there is room for improvement. Among the weaker models, there is much more headroom, with the weakest models only just equalling or surpassing the chance score. Overall, there is a large spread of performance scores across the models, suggesting the benchmark has a good range of question difficulties.

**Closed-source models are noticeably better than open-source models.** Considering the Figure→Caption task, there is a difference of 34.6% between the scores of the best closed and open-sourced models. Moreover, the best-performing open-source model, TransCore-M underperforms the worst closed-source model. This difference is more pronounced for the Caption→Figure task.

**Adversarially selected negatives are more challenging.** As an ablation, we compare model performance when answering questions with adversarially selected multiple-choice negatives and randomly selected negatives (see Tab. 2 coloured text). As expected, in the vast majority of cases, accuracy scores are higher on the random negatives – for some open-source models, the accuracy score more than doubles, and for the closed-source models, the maximum accuracy score is almost met. However, for the open-source models evaluated on the Caption→Figure task, there is almost no change in performance between the adversarial and random negative settings. Given that the scores are close to the chance score, it is likely this task is too challenging for these models.

**Caption→Figure is more difficult than Figure→Caption.** Multi-image tasks are known to be challenging to LMMs [72], and slightly higher overall scores are attained on the Figure→Caption task, especially in the random negatives setting. Considering the human baseline, a noticeably lower score is attained on the Caption→Figure task, suggesting it is easier for humans to distinguish fine-grained details in the text domain. The VLM baselines show no discernible difference in performance across the tasks, a possible reflection of their pretraining strategy of jointly aligning language and vision.

**Performance does not necessarily scale with model size.** Considering the models that we evaluate multiple checkpoint sizes (*e.g.*, IDEFICS, OmniLMM, Yi, etc.), we find that more often than not, the *smaller* model outperforms the larger checkpoint on the adversarially selected negatives, however, the opposite is true for the randomly selected negatives. Additionally, the difference in performance is more pronounced on the randomly selected negatives.

**CLIP remains a strong baseline.** Across both tasks, on questions with adversarial negatives, the CLIP baseline performs comparably or superior to the leading open-source models, though is beaten by the closed-source models. When negatives are randomly selected, CLIP far surpasses the open-source models, almost equalling GPT-4V and the Gemini-Pro models.

**Humans are a stronger baseline.** The mean human baseline outperforms all the models, though does not achieve a perfect score, reflecting the challenging nature of SciFIBench and the fact that the participants were not necessarily domain experts. As indicated by the standard deviation, a range of accuracy scores were recorded for each task, with some participants scoring equal or lower than the best LMMs. It is worth noting a caveat to the human performance is that the human verification part of the curation process could have introduced bias toward questions that are 'easier' for humans to answer.

**OCR.** As detection of fine-grained textual detail is a key component of scientific figure interpretation, it is not unexpected that an above-chance score can be attained in a text-only setting. However, given the significant difference in performance between the text-only and image settings, it is clear that interpretation of visual details is required to answer the majority of questions in our benchmark.

In the Appendix, we include qualitative results for each task and examples of model output.

## 4.4 Curated vs. noisy data

Here, we provide evidence that although our benchmark is relatively small, it is sufficiently robust. We evaluate a subset of our models on both the curated and noisy CS question sets and find that in almost every case the ranking is preserved across the datasets – and in the case where the rankings switch, the performance differential between the two models is small – suggesting there is little information to be gained by evaluating on an arbitrarily larger dataset. Additionally, we conduct bootstrapping to estimate the variance of model performance on the curated dataset. Concretely, for each task, we sample with replacement 500 times from the relevant question set and evaluate the performance of Gemini-Pro Vision (middle-performing model capable of both tasks) on the sample. Repeating this process 100k times yields a mean accuracy and variance of $(56.00, 0.05)$ and $(52.40, 0.05)$ for the two tasks. This low variance provides further evidence that our curated dataset is sufficiently representative.

| k-shot | Similar examples | Random examples ($\mu\pm$SD) |
|---|---|---|
| 0 | 56.8 | 56.8 |
| 1 | 57.6 | 55.8±1.2 |
| 2 | 61.2 | 59.4±0.6 |
| 3 | 59.2 | 58.4±1.0 |
| 4 | 58.0 | 58.3±1.0 |
| 5 | 58.0 | 58.0±1.4 |



Figure 6: Gemini-1.5 Flash **k-shot results**.

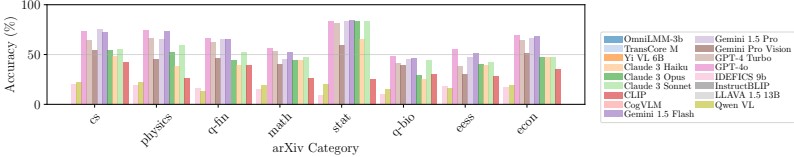

Figure 7: **Error analysis.** *Left*: per-category performance on SciFIBench. *Right*: answer refusal rate (0 indicates a valid answer for all questions).

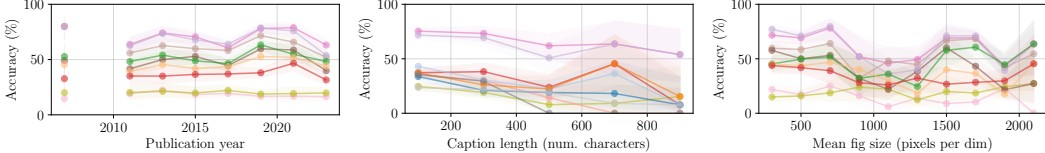

Figure 8: **Error analysis**. Left: performance with publication year for both tasks, centre: performance with caption length for the Figure→Caption task, and right: performance with mean figure size for the Caption→Figure task. Shaded regions show 95% Wilson confidence intervals.

| Model | Ranking (μ ± SD) | | | |
| --- | --- | --- | --- | --- |
| | Claude 3.5 Sonnet | GPT-4o | Gemini 1.5 Pro | **Combined** |
| GPT-4 Turbo | **2.27** ± 1.35 | **2.13** ± 1.20 | **2.07** ± 1.19 | **2.16** ± 1.12 |
| Gemini 1.5 Flash | 3.09 ± 1.64 | 2.92 ± 1.63 | 2.98 ± 1.61 | 3.00 ± 1.28 |
| Claude 3 Sonnet | 3.77 ± 2.61 | 4.14 ± 2.53 | 3.54 ± 2.31 | 3.82 ± 1.59 |
| Claude 3 Haiku | 4.00 ± 2.11 | 4.37 ± 2.25 | 4.56 ± 1.98 | 4.31 ± 1.47 |
| Gemini 1.0 Pro Vision | 4.72 ± 2.16 | 4.22 ± 1.92 | 4.33 ± 1.87 | 4.42 ± 1.42 |
| True Caption | 6.22 ± 2.03 | 5.72 ± 1.93 | 5.67 ± 2.17 | 5.87 ± 1.45 |
| Qwen VL Chat | 5.86 ± 2.03 | 5.77 ± 1.67 | 6.06 ± 1.73 | 5.90 ± 1.35 |
| TransCore-M | 7.85 ± 1.51 | 8.05 ± 1.37 | 8.09 ± 1.27 | 8.00 ± 1.18 |
| OmniLMM 3b | 8.57 ± 1.27 | 8.75 ± 1.32 | 8.83 ± 1.13 | 8.72 ± 1.12 |
| Yi VL 6b | 8.65 ± 1.53 | 8.93 ± 1.20 | 8.87 ± 1.22 | 8.82 ± 1.15 |

Table 3: **Caption generation ranking across 100 test samples**. Best ranking is 1, worst is 10.

We also analyse the degree to which the noisy data can be used to provide few-shot examples. We experimented with both 'random' and 'similar' (selected based on similarity) examples. The results for Gemini 1.5 Flash are shown in Tab.6. We find that model performance is sensitive to the prompt and with some prompt structures, the presence of examples decreases performance. However, with certain prompts as the results show, incorporating potentially noisy examples can quantitatively improve performance compared to the 0-shot setting. Additionally, a qualitative review of the outputs suggests that the model's responses more closely follow the instructed format when examples are included, which reduces the need for an LLM to parse the correct answer from the LMM output. The finding that performance can decline with increased shots in certain scenarios is consistent with other works, *e.g.* [25].

## 4.5   Error analysis

Fig.7 displays a decomposition of performance by arXiv category for the Figure→Caption task for a subset of the evaluated models. We find that the relative rankings of models remain broadly consistent across categories. However, there are clear differences in performance across the categories with most models scoring highly on 'stat' questions and much lower on 'q-bio'. Refusal rates to provide a multiple-choice answers for every question was low among the evaluated LMMs, though models with a higher proclivity for verbose outputs tended to be less decisive. We include an analysis of performance across different properties of the figure-caption pairs making up each question, including the publication year of the source arXiv paper, caption length and figure size, in Fig.8. In general, there is no clear evidence of publication year and figure size having any impact on model performance, though a slight macro-trend of decreasing performance with increasing caption length is observed.

## 4.6   Caption generation

We extend the breadth of our evaluation by assessing the capabilities of the LMMs to generate suitable captions for scientific figures. Initially, we construct a test set by randomly sampling 100 figure-caption pairs from the Figure→Caption task. We then prompt a set of test LMM baselines to generate captions for each figure and select the GPT-4o, Gemini-1.5 Pro, Claude 3.5 Sonnet [73] models to evaluate the generated captions. These specific models were chosen as evaluators because they are 'stronger' than the test models on most benchmarks and are from different model families, reducing potential bias. The generated captions were shuffled with the true caption and passed to the evaluator models to rank. We report the rankings across all samples in Tab.3. The results clearly delineate preference among the test models with the closed-source models outperforming the open-source models (as they do on SciFIBench). **Captions generated by all closed-source models are preferred over the true caption**. Conversely, the true captions are preferred over all the open-source model captions. **Strong agreement is shown between the evaluator models**. Further details can be found in the Appendix.

## 4.7   Alignment

A central motivation of this work is guiding the progress of LMMs to conduct scientific research. However, if LMMs are to be utilised as a tool for scientific acceleration, it is crucial to ensure they are aligned and the degree to which they can reliably follow instructions is known. To this end, we devise a small-scale experiment to probe this instruction-following ability and see if the models reason faithfully or are prone to 'cheating'. In addition to a control baseline, we create 4 different augmentations of the SciFIBench CS Figure→Caption questions (see Fig. 9). In two of the augmentations, we mark the true caption as <Correct>. In one of these, we additionally instruct the model to *ignore*

this extra information. For the remaining two augmentations, we repeat this process, however, we mark a randomly chosen *incorrect* caption as <Correct>. For a selection of models, we evaluate the performance accuracy on each augmented question set and display the results in Fig. 10.

**Annotating an answer as correct significantly changes performance.** We find that for all models, marking the correct answer has a noticeable increase in performance relative to the baseline. Similarly, marking the incorrect answer as correct consistently decreases the performance relative to the baseline. There are also clear differences in sensitivity to this new information. For example, the performance relative to the baseline for Qwen-VL and Gemini-Pro Vision varies at most 30%, whereas for models like LLaVA-1.5 and OmniLMM, the difference exceeds 50%.

**Some models are better at following instructions.** We can obtain a gauge of the alignment of the models by analysing the degree to which instruction to ignore the <Correct> annotation is followed. In almost every case, we find that the instruction does cause the performance to change in the desired direction (*i.e.*, towards the baseline score), though the amount of change varies depending on the model. For example, the performance of OmniLMM and TransCore-M shows almost no difference when instructed to ignore the annotation, suggesting weaker instruction-following. Whereas, the performance of CogVLM in particular changes drastically with the additional instruction.

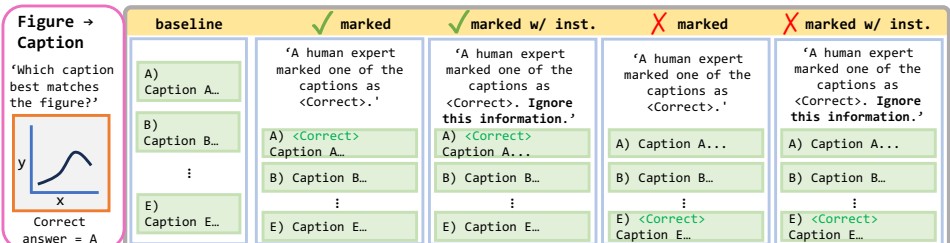

Figure 9: **Alignment experiment overview.** We create 4 augmentations of the baseline Figure→Caption questions with different information and instructions.

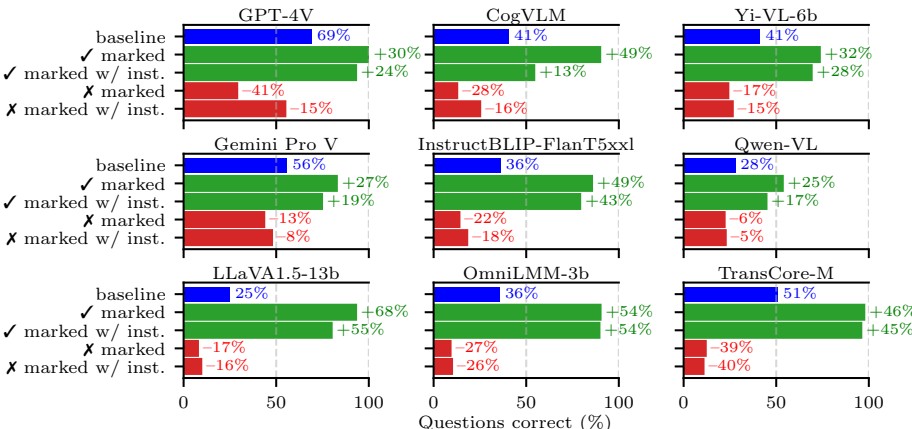

Figure 10: **Performance comparison on the augmented question sets**. Note, the labelled percentage changes reflect the change in accuracy relative to the baseline.

## 5 Conclusions

We introduce the Scientific Figure Interpretation Benchmark (SciFIBench) to evaluate the capabilities of LMMs to interpret and understand scientific figures. We curate the multiple-choice questions in our benchmark using arXiv paper figure-captions pairs from the SciCap [37] and ArXivCap [47] datasets and employ adversarial filtering to select hard negatives, increasing the difficulty of our benchmark. We use human verification when selecting questions to construct a robust, high-quality dataset that can be used to efficiently evaluate future models without the need for extensive compute or API credits. We benchmark the performance of 32 LMM, VLM and human baselines on SciFIBench, finding it to be challenging, with room for improvement. Finally, we analyse the alignment and instruction following abilities of the LMMs when answering questions in our benchmark. We release our dataset for the community to use and hope our work encourages further research in this important domain.

## Acknowledgements

This work was supported by the UKRI Centre for Doctoral Training in Application of Artificial Intelligence to the study of Environmental Risks (reference EP/S022961/1), an Isaac Newton Trust grant, a research gift from Google, an EPSRC HPC grant, the Hong Kong Research Grant Council - Early Career Scheme (Grant No. 27208022) and HKU Seed Fund for Basic Research. Samuel would like to acknowledge the support of Z. Novak and N. Novak in enabling his contribution. We thank Akash Gupta, Ansh Sharma, Arduin Findeis and Florian Langer for their valuable assistance in conducting the human baseline evaluations for our benchmark.

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

# Appendix

We structure this Appendix to our main paper into 10 parts. §A we include the datasets and benchmarks track submission checklist. §B we outline the intended uses and limitations of our SciFIBench dataset. §C we provide additional details and links to the data and code related to this project. §D we include details of how the human baseline was carried out. §E we provide additional compute details to those included in the main paper. §F we list the specific API model versions used for this work. §G we include the exact prompts used for the two tasks that make up SciFIBench. §H we provide additional quantitative experimental results, including further human baseline results, per-category results, and curated vs. noisy dataset model performance rankings. §I we demonstrate qualitative results on SciFIBench including example questions along with LMM output and reasoning, and we additionally provide examples of the LMM output before and after automatic evaluation. Finally, in §J will include examples of captions generated by the LMMs. To improve clarity, in this Appendix we format model inputs (*e.g.* prompts) as `Input` and model outputs as `Output`.


## B    Dataset intended uses and limitations

As mentioned in the main paper, our investigation focuses on our SciFIBench benchmark. The intended usage of this dataset is the evaluation of the capabilities of LMMs. None of the data contain personally identifiable information or offensive content.

In terms of limitations, we curated SciFIBench to ensure high-quality questions and answers and also feasible evaluation, however this came at the cost of volume. As such, the limited size of the benchmark means it is of limited usefulness for fine-tuning or pre-training.

## C    Data and code

### C.1    Data

The SciFIBench dataset is available via HuggingFace at this link: `https://huggingface.co/datasets/jonathan-roberts1/SciFIBench`. Following the underlying SciCap [37] and ArXiv-Cap [47] data, we give the dataset a **CC BY-NC-SA 4.0** license and bear responsibility for any license violation. The HuggingFace repository conveniently contains a dataset viewer from which the benchmark data can be viewed and downloaded. Additionally, Croissant metadata can be downloaded from this link.

### C.2    Code

We include example code and instructions for (1) interacting with the SciFIBench HuggingFace datasets object, (2) inference on both tasks using closed- and open-source models and, (3) automatic LLM evaluation of answers. These assets can be found at the following repository: `https://github.com/jonathan-roberts1/SciFIBench`. We open-source our code through an **MIT** license.

### C.3    Hosting and maintenance plan

We plan to continue hosting our dataset on HuggingFace, due to ease of integration in the datasets and transformers libraries and other useful features such as the dataset viewer. To ensure the longevity of the dataset, for the camera-ready version, we will include a DOI generated from the HuggingFace repository currently hosting SciFIBench. For the code, we provide a link to our GitHub repository containing evaluation code and other useful links.

We will continue to provide necessary maintenance to both of these repositories.

## D    Human baseline

Our human baseline was conducted on a 50-question subset of SciFIBench CS questions. The baseline was carried out via a Google form containing instructions and all the questions. All participants were given the same form that contained MCQ options in the same order as presented to the models. The form can be accessed at this link: `https://forms.gle/Rgkk4UjU2oAPbxRM8`. Participants were not compensated for taking part in our human baseline.

## E    Compute

To supplement the compute details provided in §4.2 of the main paper, we include further information here. Inference was by far the most compute-intensive element of this work, with other elements requiring negligible compute in comparison. For inference on SciFIBench, the typical open-source model required 1xA100 80GB GPU for ~45 minutes except the Emu2-Chat and IDEFICS-80b-Instruct models, which required 2xA100s.

## F    API model versions

These are the specific versions of the API models used in this work:

- GPT-4V: *gpt-4-vision-preview*
- GPT-4 Turbo: *gpt-4-turbo-2024-04-09*
- GPT-4o: *gpt-4o-2024-05-13*

- Gemini-Pro Vision: *gemini-pro-vision*
- Gemini-Pro: *gemini-1.0-pro-001*
- Gemini-Pro 1.5: *gemini-1.5-pro-preview-0409*
- Gemini-Flash 1.5: *gemini-1.5-flash-preview-0514*
- Claude 3 Opus: *claude-3-opus@20240229*
- Claude 3 Sonnet: *claude-3-sonnet@20240229*
- Claude 3 Haiku: *claude-3-haiku@20240307*

## G Prompt templates

### G.1 Task prompts

Below, we include the prompt templates used in the Figure → Caption task and Caption → Figure task. These prompt templates were utilised by each model, however, in some cases minor modifications were implemented depending on prompting suggestions mentioned by model authors. Note, items in parenthesis – *e.g.*, {*Caption*} or {*Image*} – represent the placement of the actual image or caption within the prompt.

**Figure → Caption**

```
''{Image}
A) {Caption1}
B) {Caption2}
C) {Caption3}
D) {Caption4}
E) {Caption5}
Which of the captions best describes the image?  Let's think step by step.  Only
provide the letter of the correct caption as your answer.  Answer:''
```

**Caption → Figure**

```
''A) {Image1}
B) {Image2}
C) {Image3}
D) {Image4}
E) {Image5}
{Caption}
Which of the images best matches the caption?  Let's think step by step.  Only
provide the letter of the correct image as your answer.  Answer:''
```

### G.2 Automatic evaluation

We use the following when prompting Gemini-Pro to automatically evaluate the output of all LMMs.

```
''Here is the output from a generative model:
{Model_Output}
The output contains the answer to a multiple choice question with options A) -
E). Return only the letter of the answer.  If no answer is found, return None.''
```

See Figs. 13 and 14 for examples of LMM output before and after reformatting with Gemini-Pro using the above prompt.

### G.3   Caption generation

**Generation**

> {*Figure*} ''Provide a one sentence caption that describes the figure. Do not include prefixes such as "The figure shows" or "The plot shows". Include just the caption. Caption:''

**Evaluation**

> ''Your task is to rank the following captions in order of how well they represent the figure. Figure: {*Figure*} A) <Caption A> ... J) <Caption J> Rank the captions from best to worst. Return just a list of the ranked captions, nothing else. For example: Ranked Captions: [A, B, E, F, G, J, C, I, D, H]. Ranked Captions:''

## H  Extended quantitative results

### H.1  Human baseline

The individual results for the human baseline are included below in Tab. 4. The questions were randomly sampled from the SciFIBench CS dataset (using adversarial negatives) and evaluated by 5 humans (comprising both undergraduate and postgraduate students).

| Human/Model | Accuracy | |
| --- | --- | --- |
| | Figure → Caption | Caption → Figure |
| Human 1 | **96.0** | **92.0** |
| Human 2 | 88.0 | 68.0 |
| Human 3 | 84.0 | 72.0 |
| Human 4 | 72.0 | 80.0 |
| Human 5 | 92.0 | 80.0 |
| Mean Human | 86.4 | 78.4 |
| GPT-4o | 72.0 | 76.0 |
| Gemini-Pro 1.5 | 84.0 | 72.0 |

Table 4: **Extended human baseline results.** Human baseline accuracy on a 25 question-per-task subset of SciFIBench CS dataset. The best score for each task is in **bold**. Results on the same question set for the leading LMMs (GPT-4o and Gemini-Pro 1.5) are included as a comparison.

Several observations can be drawn from these results: (1) **Human variance** – Across both tasks, there is a high degree of variance on the accuracy scores of the different humans that answered the questions. (2) **Comparison to closed-source models** – Across most axes, the ranking of the human baseline above the leading LMM baselines (GPT-4o and Gemini-Pro 1.5) outlined in the main paper is preserved. Although on each task there are some humans that are outperformed by at least one of the LMMs, the mean human beats the LMMs, which are also either beaten or equalled by the median human.

It is worth acknowledging that this comparison to the human baseline has limitations. Firstly, this is a small sample that is not necessarily representative of the entire population, yet still has considerable variance. Another consideration is the learning that could occur during the question answering. Unlike the models, which answer each question independently, the humans answered the questions as part of a survey of 50 questions (25 per task). Although feedback was not given, it is possible, for example, that exposure to earlier questions influenced the approach taken to answering later questions.

### H.2  Curated vs. noisy data rankings

To supplement the comparison of model rankings on the curated and noisy data in §4.4 of the main paper, we include Fig. 11 to illustrate how the model rankings are consistent across the datasets.

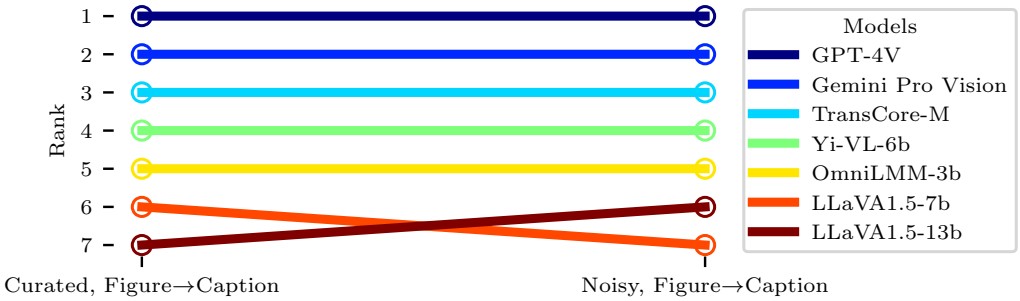

Figure 11: **Performance ranking comparison of models evaluated on the curated and noisy CS datasets.** We observe only minor variations in ranking between the datasets.

## H.3  Performance Across Categories

We analyse the performance of a representative sample of models on the Figure → Caption task across the 12 categories in the SciFIBench CS dataset. The results are displayed in Fig. 12 and comprehensive results for all models can be found in the Appendix. Although the different categories in our benchmark are all (at least partly) from the arXiv CS category, there is considerable variation in the style and type of figures within each category. This disparity is reflected in the difference in performance across categories: the average accuracy for the selected models differs by 22.5% between the best category (LG) and the worse category (AI). There is slight variation in ranking of the models across categories, but it remains fairly consistent.

In Tab. 5 below, we include full results for the per-category performance on the Figure → Caption task of the SciFIBench CS dataset. This table expands on Fig. 12 by including scores for each model displayed in the graph, as well as results for additional models evaluated.

| Model | Accuracy per category | | | | | | | | | | | | |
|---|---|---|---|---|---|---|---|---|---|---|---|---|---|
| | cs.NI | cs.CV | cs.DC | cs.SY | cs.CR | cs.AI | cs.CL | cs.RO | cs.LG | cs.SE | other cs | cross-list | **Total** |
| GPT-4V | 72.0 | 84.0 | 64.0 | 76.0 | 80.0 | 60.0 | 72.0 | 68.0 | 88.0 | 68.0 | 66.7 | 63.2 | 69.2 |
| GPT-4o | 76.0 | 84.0 | 64.0 | 80.0 | 92.0 | 20.0 | 68.0 | 76.0 | 96.0 | 72.0 | 71.2 | 78.2 | 75.4 |
| Gemini-Pro Vision | 48.0 | 76.0 | 56.0 | 56.0 | 96.0 | 10.0 | 56.0 | 32.0 | 76.0 | 64.0 | 56.1 | 48.9 | 56.0 |
| Gemini-Pro 1.5 | 76.0 | 88.0 | 52.0 | 84.0 | 76.0 | 60.0 | 60.0 | 72.0 | 92.0 | 80.0 | 72.0 | 75.2 | 74.2 |
| Claude 3 Haiku | 48.0 | 56.0 | 44.0 | 60.0 | 76.0 | 20.0 | 64.0 | 64.0 | 48.0 | 56.0 | 52.3 | 47.4 | 52.6 |
| Claude 3 Sonnet | 64.0 | 68.0 | 40.0 | 64.0 | 76.0 | 60.0 | 48.0 | 52.0 | 68.0 | 60.0 | 46.2 | 48.9 | 53.4 |
| Claude 3 Opus | 56.0 | 60.0 | 44.0 | 64.0 | 80.0 | 30.0 | 64.0 | 68.0 | 76.0 | 56.0 | 59.8 | 57.1 | 60.0 |
| IDEFICS-9b-Instruct | 4.0 | 32.0 | 24.0 | 20.0 | 12.0 | 20.0 | 16.0 | 24.0 | 44.0 | 12.0 | 18.9 | 21.8 | 20.6 |
| IDEFICS-80b-Instruct | 16.0 | 28.0 | 20.0 | 24.0 | 20.0 | 20.0 | 0.0 | 12.0 | 4.0 | 20.0 | 28.0 | 21.1 | 20.6 |
| Qwen-VL-Chat | 20.0 | 20.0 | 20.0 | 28.0 | 52.0 | 20.0 | 40.0 | 20.0 | 28.0 | 32.0 | 30.3 | 24.8 | 28.0 |
| Emu2-Chat | 12.0 | 32.0 | 20.0 | 16.0 | 40.0 | 20.0 | 20.0 | 8.0 | 20.0 | 36.0 | 21.2 | 17.3 | 20.8 |
| TransCore-M | 48.0 | 52.0 | 44.0 | 44.0 | 64.0 | 30.0 | 56.0 | 32.0 | 60.0 | 48.0 | 57.6 | 48.1 | 51.0 |
| InternLM-XComposer-7b | 32.0 | 40.0 | 44.0 | 40.0 | 32.0 | 50.0 | 40.0 | 32.0 | 44.0 | 36.0 | 25.8 | 34.6 | 34.0 |
| InternLM-XComposer2-7b | 16.0 | 24.0 | 20.0 | 28.0 | 40.0 | 20.0 | 20.0 | 36.0 | 36.0 | 40.0 | 40.9 | 14.3 | 28.0 |
| CogVLM-Chat | 40.0 | 32.0 | 48.0 | 40.0 | 44.0 | 20.0 | 40.0 | 40.0 | 56.0 | 52.0 | 40.2 | 38.3 | 40.8 |
| OmniLMM-3b | 36.0 | 32.0 | 24.0 | 24.0 | 36.0 | 20.0 | 60.0 | 40.0 | 56.0 | 20.0 | 36.4 | 35.3 | 35.8 |
| OmniLMM-12b | 40.0 | 40.0 | 32.0 | 40.0 | 56.0 | 20.0 | 52.0 | 32.0 | 40.0 | 36.0 | 34.8 | 23.3 | 34.2 |
| Yi-VL-6b | 44.0 | 36.0 | 36.0 | 24.0 | 60.0 | 40.0 | 36.0 | 44.0 | 64.0 | 44.0 | 40.9 | 39.1 | 41.4 |
| Yi-VL-34b | 32.0 | 44.0 | 20.0 | 20.0 | 48.0 | 10.0 | 32.0 | 28.0 | 60.0 | 40.0 | 31.1 | 30.1 | 32.6 |
| InstructBLIP-FlanT5-xl | 32.0 | 28.0 | 32.0 | 48.0 | 40.0 | 50.0 | 44.0 | 48.0 | 36.0 | 40.0 | 32.6 | 33.1 | 35.8 |
| InstructBLIP-FlanT5-xxl | 60.0 | 32.0 | 36.0 | 32.0 | 40.0 | 30.0 | 56.0 | 36.0 | 40.0 | 40.0 | 31.8 | 32.3 | 36.2 |
| InstructBLIP-Vicuna-7b | 16.0 | 32.0 | 20.0 | 28.0 | 24.0 | 30.0 | 12.0 | 28.0 | 24.0 | 16.0 | 18.9 | 20.3 | 21.0 |
| InstructBLIP-Vicuna-13b | 24.0 | 16.0 | 24.0 | 24.0 | 36.0 | 10.0 | 28.0 | 16.0 | 8.0 | 28.0 | 21.2 | 23.3 | 22.2 |
| Monkey-Chat | 16.0 | 20.0 | 12.0 | 28.0 | 56.0 | 20.0 | 48.0 | 20.0 | 16.0 | 28.0 | 32.6 | 22.6 | 27.2 |
| LLaVA-1.5-7b | 20.0 | 32.0 | 44.0 | 32.0 | 56.0 | 10.0 | 48.0 | 28.0 | 32.0 | 12.0 | 30.3 | 35.3 | 35.8 |
| LLaVA-1.5-13b | 20.0 | 28.0 | 16.0 | 16.0 | 44.0 | 20.0 | 32.0 | 16.0 | 28.0 | 28.0 | 25.8 | 24.1 | 25.0 |
| CLIP ViT-H-14-378-quickgelu | 40.0 | 52.0 | 32.0 | 32.0 | 52.0 | 20.0 | 64.0 | 32.0 | 64.0 | 32.0 | 40.2 | 40.6 | 41.8 |
| MetaCLIP ViT-H-14-quickgelu | 28.0 | 44.0 | 20.0 | 32.0 | 48.0 | 10.0 | 56.0 | 48.0 | 32.0 | 44.0 | 39.4 | 31.6 | 36.6 |
| Google Multimodal Embedding | 36.0 | 72.0 | 16.0 | 36.0 | 56.0 | 20.0 | 76.0 | 56.0 | 68.0 | 60.0 | 44.7 | 43.6 | 47.6 |
| **Total** | 37.0 | 44.3 | 33.7 | 39.3 | 52.8 | 26.6 | 45.0 | 38.1 | 48.6 | 41.7 | 39.6 | 37.1 | 39.9 |

Table 5: **Full-per category performance on the SciFIBench CS dataset, Figure → Caption task**.

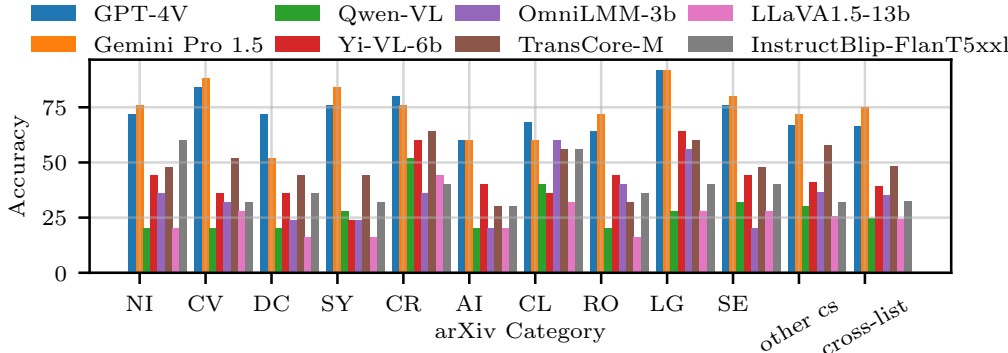

Figure 12: **Performance variation across question categories** for selected models on the Figure → Caption task of the CS dataset.

# I  Qualitative results

In this section, we present qualitative results to complement the quantitative results included in the main paper and show additional example SciFIBench questions. We structure these results in the following way: (1) **Automatic postprocessing** – For selected questions, we provide examples (Figs. 13-14) of both the raw output from a selection of the LMMs evaluated, and the formatted output following automatic answer extraction by Gemini-Pro. (2) **Model reasoning** – For a number of example questions (Figs. 15-18), we provide the reasoning steps taken by the models when producing an answer.

## I.1  Automatic postprocessing

In Figs. 13 and 14 we provide examples of LMM output before and after automatic evaluation by Gemini-Pro. The examples show that Gemini-Pro is able to extract the correct prediction from the unformatted LMM output, greatly aiding automatic evaluation. For most models, only minor output formatting is required as the correct letter answer is given at the start of the output. However in some cases, such as the output of CogVLM in Fig 14, which begins: '*The correct caption that best describes the image is:...*', this is not the case and more involved reasoning is required to extract the correct answer choice.

**Figure → Caption**: Which of the captions best describes the image?

Image:

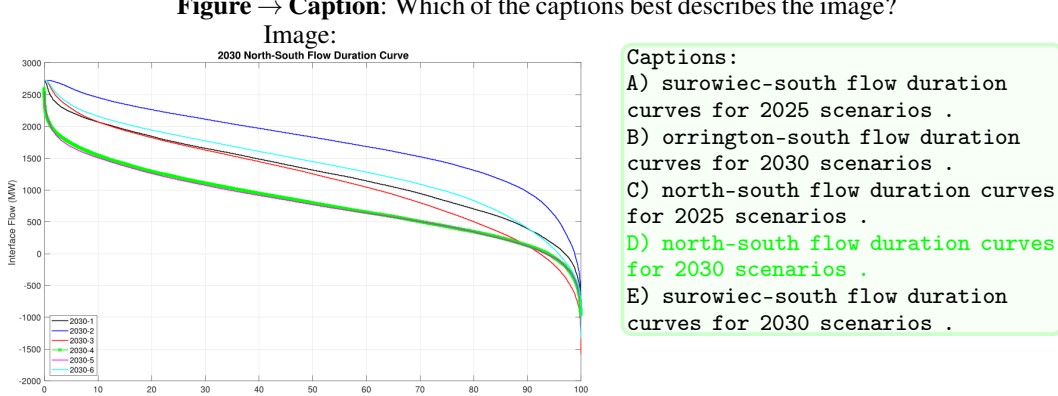

Captions:
A) surowiec-south flow duration curves for 2025 scenarios .
B) orrington-south flow duration curves for 2030 scenarios .
C) north-south flow duration curves for 2025 scenarios .
D) north-south flow duration curves for 2030 scenarios .
E) surowiec-south flow duration curves for 2030 scenarios .

**Model outputs:**

| Model | Raw output | Gemini-Pro output | Answer correct? |
|---|---|---|---|
| GPT-4V [5] | D) | D | ✓ |
| Gemini-Pro Vision [7] | D | D | ✓ |
| CogVLM [58] | D) north-south flow duration curves for 2030 scenarios. | D | ✓ |
| Yi-VL-6B [60] | C | C | ✗ |
| Qwen-VL [6] | A) surowiec-south flow duration curves for 2025 scenarios . | A | ✗ |
| OmniLMM-3B [59] | C | C | ✗ |
| TransCore-M [56] | A) surowiec-south flow duration curves for 2025 scenarios . | A | ✗ |
| LLaVA-1.5-13b [11] | C) north-south flow duration curves for 2025 scenarios . | C | ✗ |

Figure 13: **Example of automatic output evaluation by Gemini-Pro**. The example includes: (1) a randomly selected question from the SciFIBench CS dataset, Figure → Caption task; and, (2) Example LMM outputs before (raw) and after answer extraction by Gemini-Pro. [Note, the correct answer is D)].

**Figure → Caption**: Which of the captions best describes the image?
Image:

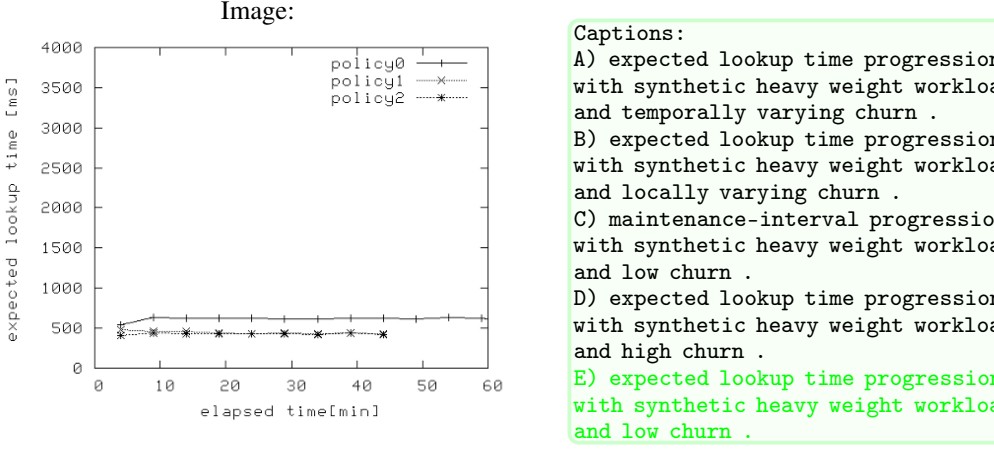

Captions:
A) expected lookup time progression with synthetic heavy weight workload and temporally varying churn .
B) expected lookup time progression with synthetic heavy weight workload and locally varying churn .
C) maintenance-interval progression with synthetic heavy weight workload and low churn .
D) expected lookup time progression with synthetic heavy weight workload and high churn .
E) expected lookup time progression with synthetic heavy weight workload and low churn .

**Model outputs:**

| Model | Raw output | Gemini-Pro output | Answer correct? |
|---|---|---|---|
| GPT-4V [5] | E) | E | ✓ |
| Gemini-Pro Vision [7] | A | A | ✗ |
| CogVLM [58] | The correct caption that best describes the image is: E) expected lookup time progression with synthetic heavy weight workload and low churn. | E | ✓ |
| Yi-VL-6B [60] | E | E | ✓ |
| Qwen-VL [6] | A) expected lookup time progression with synthetic heavy weight workload and temporally varying churn . | A | ✗ |
| OmniLMM-3B [59] | B | B | ✗ |
| TransCore-M [56] | A) expected lookup time progression with synthetic heavy weight workload and temporally varying churn . | A | ✗ |
| LLaVA-1.5-13b [11] | D) expected lookup time progression with synthetic heavy weight workload and high churn . | D | ✗ |

Figure 14: **Another example of automatic output evaluation by Gemini-Pro**. The example includes: (1) a randomly selected question from the SciFIBench CS dataset, Figure → Caption task; and, (2) Example LMM outputs before (raw) and after answer extraction by Gemini-Pro. [Note, the correct answer is E)].

## I.2    Model reasoning

In the following subsection, we provide a series of qualitative examples of LMM reasoning when answering questions from the SciFIBench CS dataset. Specifically, we amend the prompts given in Sec. G to the following:

**Figure → Caption**

```
''{Image}
A) {Caption1}
B) {Caption2}
C) {Caption3}
D) {Caption4}
E) {Caption5}
Which of the captions best describes the image?  Let's think step by step.
Answer:''
```

**Caption → Figure**

```
''A) {Image1}
B) {Image2}
C) {Image3}
D) {Image4}
E) {Image5}
{Caption}
Which of the images best matches the caption?  Let's think step by step.
Answer:''
```

*i.e.*, allowing the models to outline their reasoning steps by removing the output format constraint:

```
Only provide the letter of the correct caption as your answer.
```

We structure each of the following examples as (i) **Question** – showing the question figure(s) and caption(s); and (ii) **Outputs** – showing the outputs from a selection of the LMMs evaluated. We provide examples for the Figure → Caption task (Figs. 15-16) and for the Caption → Figure task (Figs. 17-18).

**Example 1: Figure → Caption** – Which of the captions best describes the figure?

Figure:

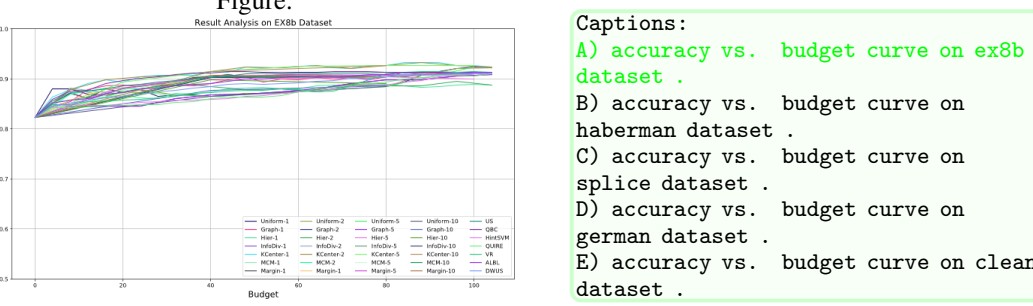

```
Captions:
A) accuracy vs. budget curve on ex8b
dataset .
B) accuracy vs. budget curve on
haberman dataset .
C) accuracy vs. budget curve on
splice dataset .
D) accuracy vs. budget curve on
german dataset .
E) accuracy vs. budget curve on clean
dataset .
```

Figure 15: Example Figure → Caption question. [Note, the correct answer is A].

**Model outputs:**

*GPT-4V*:

```
The image shows a graph with multiple lines representing different methods or
algorithms, each plotted against two axes: "Budget" on the x-axis and "Accuracy"
on the y-axis. The title of the graph is "Result Analysis on EX8b Dataset."
Given the options provided:
A) accuracy vs. budget curve on ex8b dataset. B) accuracy vs. budget curve on
haberman dataset. C) accuracy vs. budget curve on splice dataset. D) accuracy vs.
budget curve on german dataset. E) accuracy vs. budget curve on clean dataset.
The caption that best describes the image is:
A) accuracy vs. budget curve on ex8b dataset.
This is because the title of the graph explicitly mentions the "EX8b Dataset," which
matches the dataset mentioned in option A.
```

*Gemini-Pro Vision*:

```
The image shows multiple lines that track the accuracy of different algorithms as
the budget increases. The caption describes the contents of the image well. So the
answer is: A) accuracy vs. budget curve on ex8b dataset.
```

*Qwen VL*:

```
The image shows a graph with "Accuracy" on the y-axis and "Budget" on the x-axis. The
graph represents the result analysis on the EX8b dataset.
Therefore, the captions that best describe the image are:
A) Accuracy vs. Budget curve on EX8b dataset B) Accuracy vs. Budget curve on
Haberman dataset C) Accuracy vs. Budget curve on Splice dataset D) Accuracy vs.
Budget curve on German dataset E) Accuracy vs. Budget curve on Clean dataset
```

*OmniLMM-3B*:

```
A) accuracy vs. budget curve on ex8b dataset
```

*TransCore-M*:

```
The image presents a graph titled "Retrieve Analysis on EXB Dataset". The x-axis of
the graph is labeled "Accuracy" and the y-axis is labeled "Budget". The graph shows
a series of curves, each representing a different dataset. The curves are colored
differently, with blue, green, purple, and orange lines. The blue line represents
the accuracy vs. budget curve on the EXB dataset, the green line represents the
accuracy vs. budget curve on the Haberman dataset, the purple line represents the
accuracy vs. budget curve on the Splice dataset, and the orange line represents
the accuracy vs. budget curve on the German dataset. The graph provides a visual
representation of the relationship between accuracy and budget for each dataset.
```

*LLaVA-1.5-13b*:

The image shows a graph with a curve that represents the accuracy of a machine learning model as a function of the budget. The graph is labeled with the dataset name, which is "ex8b". The curve demonstrates the trade-off between accuracy and budget, indicating that increasing the budget can lead to better accuracy.

*Yi-VL-6b*:

A) accuracy vs. budget curve on ex8b dataset

*IDEFICS-9b-instruct*:

Answer: E

*CogVLM*:

The correct option is C) accuracy vs. budget curve on splice dataset. This is because the image specifically mentions 'splice dataset' in its title and the x-axis represents the 'Budget'.

Figure:

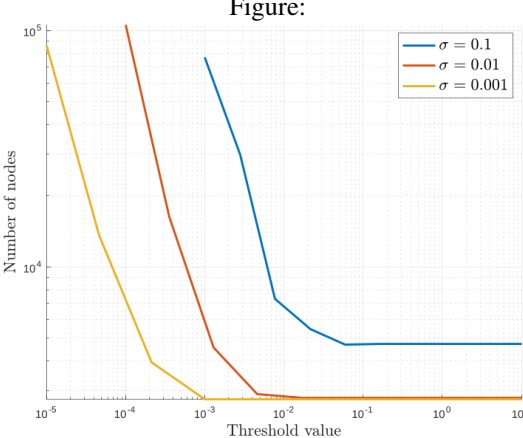

Captions:
A) number of explored nodes for randomly generated problems .
B) computation times for randomly generated problems .
C) number of explored nodes for randomly generated problems without queue algorithm .
D) computation times for randomly generated problems as a function of threshold $\delta$ .
E) number of explored nodes for randomly generated problems as a function of threshold $\delta$ .

Figure 16: Example Figure → Caption question. [Note, the correct answer is E)].

## Model outputs:

*GPT-4V*:

```
To determine which caption best describes the image, let's analyze the elements
present in the graph:
1.   The x-axis is labeled "Threshold value," which is on a logarithmic scale.  2.
The y-axis is labeled "Number of nodes," which is also on a logarithmic scale.  3.
There are three lines on the graph, each representing different values of σ (0.1,
0.01, 0.001).
Given this information, the graph is showing the relationship between the "Threshold
value" and the "Number of nodes" for different σ values.  It does not provide any
information about computation times or the use of a queue algorithm.
Therefore, the caption that best describes the image is:
E) number of explored nodes for randomly generated problems as a function of
threshold δ.
This caption correctly identifies that the graph is showing the number of explored
nodes (y-axis) as a function of the threshold value (x-axis) for different σ values.
```

*Gemini-Pro Vision*:

```
The image shows the number of nodes as a function of the threshold value for three
different values of the standard deviation.  The caption that best describes the
image is "Number of explored nodes for randomly generated problems as a function of
threshold δ."
```

*Qwen VL*:

```
The image shows a graph with several lines representing different scenarios.  The
x-axis represents the number of nodes explored and the y-axis represents the
threshold value.  There are four scenarios shown in the graph:  A) Number of explored
nodes for randomly generated problems, B) Computation times for randomly generated
problems, C) Number of explored nodes for randomly generated problems without queue
algorithm, and D) Computation times for randomly generated problems as a function of
threshold δ.
```

*OmniLMM-3B*:

```
E) number of explored nodes for randomly generated problems as a functionof threshold
δ .
```

*TransCore-M*:

The image presents a graph that illustrates the relationship between the number of
explored nodes and the computation time for randomly generated problems.  The x-axis
of the graph represents the number of explored nodes, while the y-axis represents
the computation time.  The graph is divided into three distinct sections, each
corresponding to a different threshold value.
The first section, labeled "A", shows a linear relationship between the number
of explored nodes and the computation time.  This suggests that as the number of
explored nodes increases, the computation time also increases linearly.
The second section, labeled "B", shows a quadratic relationship between the number
of explored nodes and the computation time.  This indicates that as the number of
explored nodes increases, the computation time increases at a faster rate than in the
first section.
The third section, labeled "C", shows a linear relationship between the number of
explored nodes and the computation time.  This is similar to the first section,
suggesting that the computation time increases linearly with the number of explored
nodes.
The graph also includes a legend that explains the different colors used to represent
the different threshold values.  The legend states that the blue line represents the
computation time for randomly generated problems, the orange line represents the
computation time for randomly generated problems without queue algorithm, and the red
line represents the computation time for randomly generated problems as a function of
threshold $\delta$.
Overall, the graph provides a clear visual representation of the impact of different
threshold values on the computation time for randomly generated problems.

*LLaVA-1.5-13b*:

The image shows a graph with a blue line and an orange line.  The blue line represents
the number of explored nodes for randomly generated problems, while the orange line
represents the computation times for randomly generated problems.  The graph is
labeled "threshold value" and is likely used to analyze the performance of a queue
algorithm.

*Yi-VL-6b*:

D

*IDEFICS-9b-instruct*:

Answer:  C

*CogVLM*:

The caption that best describes the image is:  E) number of explored nodes for
randomly generated problems as a function of threshold $\delta$.

**Example 3: Caption → Figure** – Which of the figures best describes the caption?

Caption: rank-25 cmc plot .

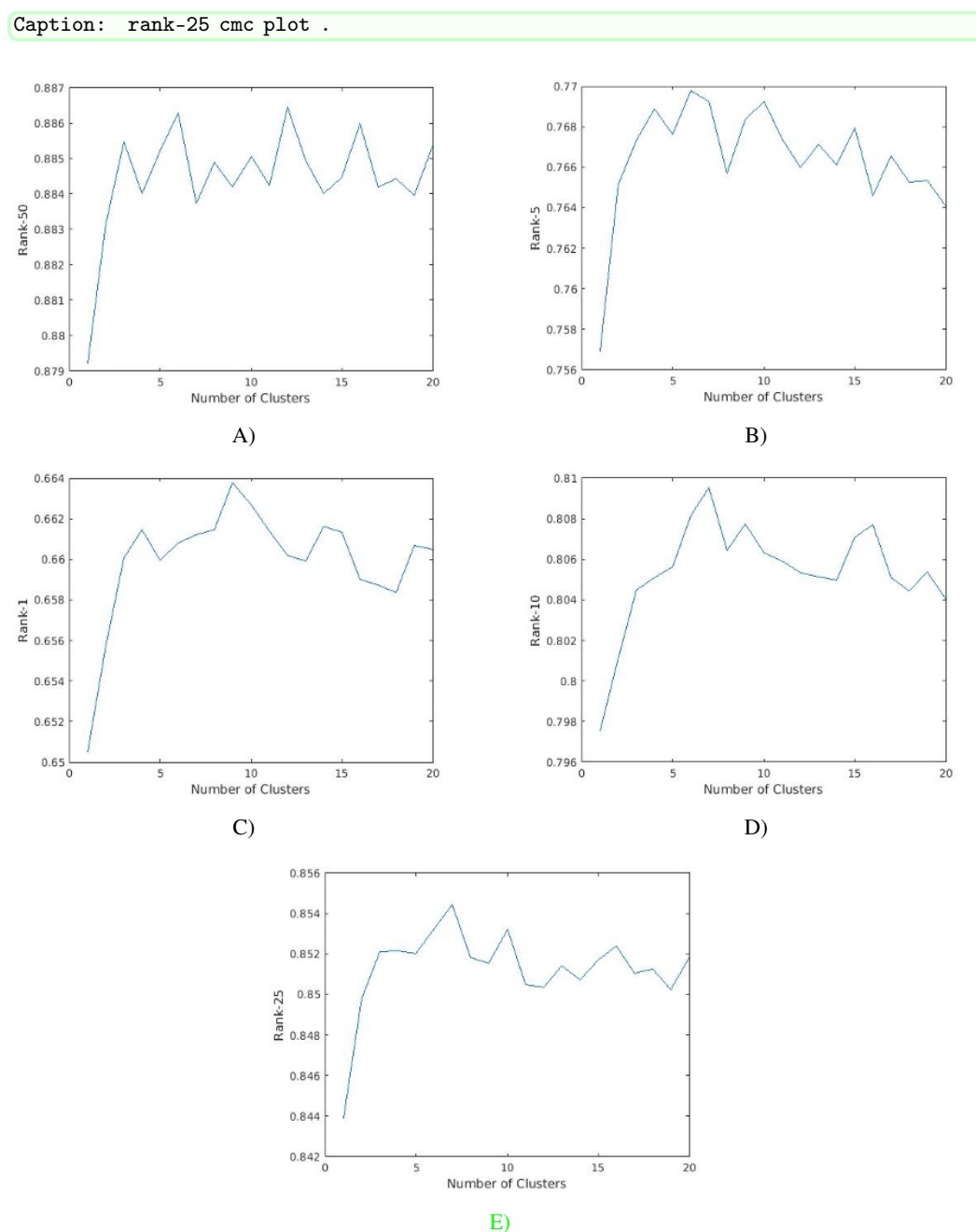

Figure 17: Example Caption → Figure question. [Note, the correct answer is E)].

**Model outputs:**

*GPT-4V:*

The caption provided is "rank-25 cmc plot." To match this caption with the correct
image, we need to look for a plot that is labeled with "Rank-25" on the y-axis and
"Number of Clusters" on the x-axis.
Let's examine each image:
A) The y-axis is labeled "Rank-50," so this does not match the caption.  B) The y-axis
is labeled "Rank-5," so this does not match the caption.  C) The y-axis is labeled
"Rank-1," so this does not match the caption.  D) The y-axis is labeled "Rank-10," so
this does not match the caption.  E) The y-axis is labeled "Rank-25," which matches
the caption, and the x-axis is labeled "Number of Clusters."
Based on the provided caption, the image that best matches is:
E) The plot with "Rank-25" on the y-axis and "Number of Clusters" on the x-axis.

*Gemini-Pro Vision*:

The caption is "rank-25 cmc plot".  The image that best matches the caption is E.
The x-axis of the image is "Number of Clusters", and the y-axis is "Rank-25".  The
caption is about the relationship between the number of clusters and the rank-25
cmc score.  The image shows that the rank-25 cmc score increases as the number of
clusters increases.

**Example 4: Caption → Figure** – Which of the figures best describes the caption?

Caption:   control signal of 2p for step input and load disturbance with istes based tuning .

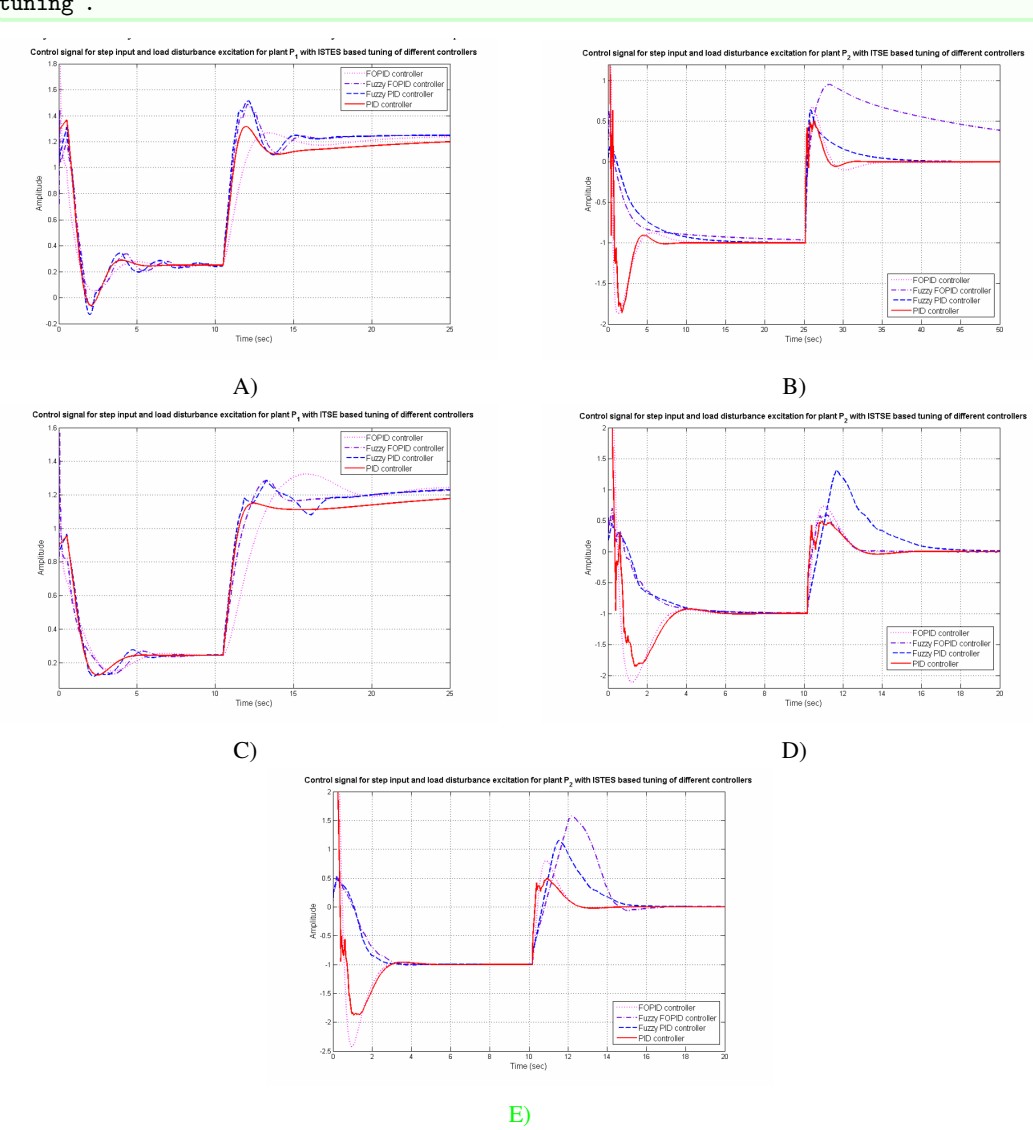

A)

B)

C)

D)

E)

Figure 18: Example Caption → Figure question. [Note, the correct answer is E)].

**Model outputs:**

*GPT-4V*:

To find the image that best matches the caption, we need to look for a few key
elements mentioned in the caption:
1.  The caption refers to a "control signal for step input and load disturbance." 2.
It mentions "plant P2," which suggests that we are looking for a graph related to a
system or process labeled as "P2." 3.  The caption also specifies "ISTES based tuning
of different controllers."
Now, let's examine each image:
A) The title of this graph refers to "plant P1," which does not match the "plant P2"
mentioned in the caption.
B) This graph's title correctly refers to "plant P2," and it also mentions "ITSE
based tuning of different controllers," which is similar to "ISTES based tuning"
in the caption.  However, there might be a typo in the caption, as "ISTES" is not a
common term in control systems, whereas "ITSE" stands for Integral of Time-weighted
Squared Error, which is a performance criterion for tuning controllers.
C) The title of this graph refers to "plant P1," which does not match the "plant P2"
mentioned in the caption.
D) This graph's title correctly refers to "plant P2," and it also mentions "ITSTE
based tuning of different controllers," which could be a typo or variation of "ISTES
based tuning" mentioned in the caption.
E) This graph's title correctly refers to "plant P2," and it also mentions "ISTES
based tuning of different controllers," which exactly matches the caption.
Based on the information provided in the caption and the titles of the graphs, the
image that best matches the caption is:
E) This graph's title matches the caption exactly, referring to "plant P2" and "ISTES
based tuning of different controllers." Despite the potential typo in the caption,
this image is the closest match.

*Gemini-Pro Vision*:

Image D best matches the caption.  The caption states that the image is of the control
signal of 2p for step input and load disturbance with istes based tuning.  The image
shows three lines, which are likely the control signals for the three different
controllers.  The lines are labeled "FOPID controller", "Fuzzy PID controller", and
"PID controller".  The image also has a legend, which shows that the x-axis is time in
seconds and the y-axis is amplitude.

## J   Generated Captions

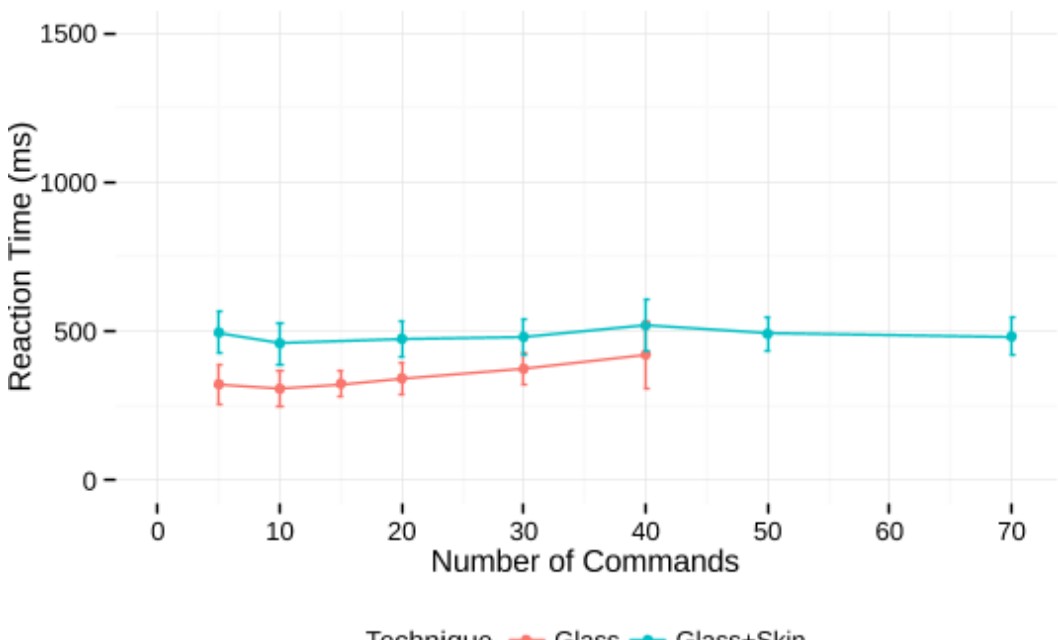

Figure 19: Example figure from the SciFIBench CS set.

True Caption:

```
mean rt vs.  the number of commands for each condition .
```

Generated Captions:

```
Claude 3 Haiku:
The graph shows the relationship between the number of commands and the reaction time
for three different techniques:  Technique, Glass, and Glass+Skin.
Claude 3 Sonnet:
Comparison of reaction times for different techniques (Glass and Glass+Skin) across
varying numbers of commands.
Gemini Pro Vision:
Reaction time as a function of the number of commands for the glass and glass+skin
conditions.
Gemini 1.5 Flash:
Reaction time increases with the number of commands, but the increase is less
pronounced for the Glass+Skin technique.
GPT-4 Turbo:
Reaction time in milliseconds for two techniques, Glass and Glass+Skin, across
varying numbers of commands.
OmniLMM-3b:
The line graph shows the relationship between glass and skin.
Qwen VL:
The reaction time of the Glass and Glass+Skin techniques as a function of the number
of commands
TransCoreM:
A graph showing the number of commands for different techniques.
Yi VL 6b:
A graph of reaction time vs number of commands shows a linear relationship.
```

