# OpenReview forum: "SciFIBench: Benchmarking Large Multimodal Models for Scientific Figure Interpretation"
_NeurIPS.cc/2024/Datasets_and_Benchmarks_Track — NeurIPS 2024 Track Datasets and Benchmarks Poster_

### Official Review · Reviewer_8auP · 2024-07-22
**Review of SciFIBench**

**Rating:** 7
**Confidence:** 3
**Correctness:** Claims are correct.
**Clarity:** The paper is well-written

**Review:**

The introduced benchmark is novel as a test suite for the new multi-modal LLM. The benchmark is well-constructed, and the authors present the method and quality assurance of the data well. Experiments are extensive, and further adversarial experiments are also conducted to check the capabilities of different multi-modal LLMs. However, the task of “Scientific figure interpretation” may go beyond the relation of figures and captions in scientific papers, and further extension of the approach may be needed to evaluate this challenging task properly.

**Strengths:**

The work is good compared with existing work related to the task.

Experiments with the benchmark are produced in a wide range of LLM, and additional adversarial experiments to check the trustworthiness of the paper are also produced.

Further experiments are produced to show the robustness of the benchmark's gold standard proposed by the authors.

The curation process and the experimental setup design are clearly explained.

**Additional Feedback:**

While captions are very helpful for understanding scientific figures, some of these are also explained in the running text of scientific papers.  A benchmark for scientific figure interpretation would also need to be added to this information to provide proper context to the LLMs.

**Documentation:**

An extensive annex is provided explaining the experiments and the benchmark. Data is provided together with the publication.

**Ethics:**

There is no evident ethical issue.

**Limitations:**

The proposed task itself has some limitations in its design. Conceptually, scientific figure interpretation goes beyond the relation between figures and captions. While captions are a good source of figure descriptions, many do not contain all the relevant information about the figure.

**Opportunities For Improvement:**

Scientific figure interpretation could go beyond the relation between captions and figures. While captions are very helpful for understanding scientific figures, some of these are also explained in the running text of scientific papers.  A benchmark for scientific figure interpretation would also need to be added to this information to provide proper context to the LLMs.

**Relation To Prior Work:**

The paper is well related with prior work.

**Summary And Contributions:**

The paper introduces a new benchmark for scientific figure interpretation. The benchmark is composed of 1000 human-curated figure-caption pairs extracted from arXiv papers, with multiple-choice questions for guessing which caption corresponds to the figure and which figure corresponds to the caption. It also provides another two subsets with much bigger Figure-Pair captions not curated by humans for downstream fine-tuning and few-shot examples.

The benchmark is then tested with 27 LLM, both open-source and closed-source. The experiment setup is presented, followed by the results of the two proposed tasks (figure-caption and caption-figure). The experiment is presented along the human evaluation baseline. GPT-4o and Gemini-Pro 1.5 Pro are the best-performing models in the benchmark below the human evaluation baseline. In addition, the authors conducted other experiments, adding incorrect information to the prompts to check the performance variability of the model and their ability to follow the instructions correctly.

---

> ### Author Rebuttal · Authors · 2024-08-16
>
> Thank you very much for taking the time to review our paper and providing suggestions for improvements. We are pleased that you consider our benchmark to be well-constructed, our experiments extensive and their design clearly explained. In the following, we include responses to the concerns raised in your review.
>
> **Adding textual context to questions**: That’s a good suggestion, and something that is done in other works (e.g., https://arxiv.org/abs/2306.03491). For SciFIBench, we wanted to focus on understanding and interpretation with just the captions (when reading research papers, people are typically drawn to figures and at least initially view them in isolation to the text). By utilising human-in-the-loop as part of our curation strategy, we ensured there was sufficient information in the captions to determine the relevant caption/figure - we will highlight this in our final version. However, exploring the addition of accompanying contextual information would be an interesting follow-up.

---

> > ### Author Rebuttal · Authors · 2024-08-31
> >
> > Thank you again for your time reviewing our work. At the top of this page we have added a summary of our combined rebuttal to all the reviewers.

---

### Official Review · Reviewer_9Ksv · 2024-07-23
**A good multiple-choice QA dataset for scientific figures in literature.**

**Rating:** 6
**Confidence:** 4
**Clarity:** The paper is well-written and easy to…

**Review:**

* Overall, the quality and clarity of the presented dataset in the draft are reasonable.
* The originality is limited. The datasets are built upon the existing benchmark dataset SciCap, and the dataset construction process follows prior work. No new methods are proposed.
* The significance of this work could be high, given the growth of scientific literature and LLM development. The dataset could benefit AI for science applications.

**Strengths:**

* Target an important domain: scientific figure understanding, which is potentially impactful to inspire the following model work for scientific literature understanding
* Dataset demos, including the Hugging Face portal and Github link, are well-prepared
* Comprehensive LLM benchmarking

**Additional Feedback:**

I suggest that the authors include the Huggingface dataset link and the GitHub dataset link in the main body of the draft. Currently, it is in the supplementary materials.

**Correctness:**

Methods and experiment designs are generally reasonable.
Minor:
- Line 49 says 26 LLMs, Line 57 says 27 LLMs

**Documentation:**

Overall, the documentation and reproducibility of this work is satisfactory
This work has a public Huggingface demo for the gold dataset and a GitHub for large-scale brand and silver datasets. I like this presentation format, which is easy for the community to gain an overview of the dataset and gain access to it.

**Ethics:**

I think there are no ethical concerns.

**Limitations:**

Besides what is mentioned above, I list some others below:
* The work should include detailed statistics and distribution of dataset domains/categories, types of figures (e.g., line chart, bar chart), and captions complexity (e.g., length, word usage, etc)

**Opportunities For Improvement:**

* Include more generic and complex scenarios for evaluating LLM capabilities of understanding scientific figures: captions and insights generation for scientific figures
* Improve the scale of the benchmark dataset

**Relation To Prior Work:**

This work is well-contextualized within the related work

**Summary And Contributions:**

This paper contributes a multiple-choice QA dataset for scientific figure understanding. The dataset is curated based on prior work SciCap, featuring an adversarial filtering approach, and consists of `figure -> caption` and `caption -> figure` tasks. The dataset is evaluated on open-sourced and closed-sourced LLMs. The evaluation results reveal the limitations in current LLMs.

---

> ### Author Rebuttal · Authors · 2024-08-16
>
> Thank you for taking the time to review our paper and providing helpful feedback. We are pleased you found our benchmarking comprehensive, targeting an important domain and thought our dataset demos well-prepared. In the following, we include responses to the concerns raised in your review.
>
> **Include more generic and complex scenarios (e.g., captions generation)**: Thank you for this suggestion. To address it, we propose taking a sample of figures from the SciFIBench gold set and prompting a selection of models to generate corresponding captions. We will then evaluate which models produce the best captions by using a set of holdout LMMs as judges. Then, we can compare the generated captions to the true captions. We plan to provide results for this prior to the end of the reviewer/author discussion period.
>
> **Improve the scale of the dataset**: Thank you for raising the question of dataset scale. While we deliberately kept the scale of SciFIBench relatively small (to enable efficient evaluation), we see benefits in increasing the scale, especially to include more categories (as raised by Reviewer X2MV). To address this, we propose to expand SciFIBench by leveraging the ArxivCap data (https://huggingface.co/datasets/MMInstruction/ArxivCap), focusing on the arXiv categories not included in SciFIBench. We will construct the questions using the methodology outlined in the SciFIBench paper. We aim to provide preliminary results for questions from these new categories before the end of the reviewer/author discussion period.
>
> **Additional dataset statistics**: We include pie charts with the category distributions for each SciFIBench task in our main paper and will add detailed statistics for the types of figure and caption complexity distribution in the final version.
>
> **Number of LLMs typo**: Good spot, we will amend this in the final version.
>
> **HuggingFace and GitHub links**: Thanks, we will add these to the final paper.

---

> > ### Author Rebuttal · Authors · 2024-08-30
> >
> > **Captions generation:**
> >
> > Following your suggestion, we have carried out the abovementioned captions generation experiment. Here is a summary of our methodology and results.
> >
> > 1. Randomly sample 100 figure-caption pairs from the Figure->Caption task.
> > 2. Select a set of (9) test LMMs:
> > {‘Claude-3 Haiku’, ‘Claude-3 Sonnet’, ‘Gemini-1.0 Pro Vision’, ‘Gemini-1.5 Flash’, ‘GPT-4 Turbo’, OmniLMM-3b’, ‘Qwen VL Chat’, ‘TransCore-M’, ‘Yi VL 6b’}.
> > 3. For each model, generate captions for each figure using the prompt:
> >
> > *Provide a one sentence caption that describes the figure. Do not include prefixes such as "The figure shows" or "The plot shows". Include just the caption. Caption:*
> >
> > 4. Define a set of strong evaluator LMMs:
> > {‘GPT-4o’, ‘Gemini-1.5 Pro’, ‘Claude 3.5 Sonnet’}. Note: each evaluator LMM is ‘stronger’ than the test models on most benchmarks.
> >
> > 5. Shuffle the generated captions with the true caption for a total of 10 captions per test figure.
> >
> > 6. For each evaluator model, rank the captions using the following prompt:
> >
> > *Your task is to rank the following captions in order of how well they represent the figure.
> > Figure: <figure>
> > A) <Caption A>
> > …
> > J) <Caption J>
> > Rank the captions from best to worst. Return just a list of the ranked captions, nothing else. For example: Ranked Captions: [A, B, E, F, G, J, C, I, D, H]. Ranked Captions:*
> >
> > 7. Compute average per-evaluator average rankings and standard deviation across all 100 samples
> >
> > 8. Compute macro-mean (µ) and standard deviation (SD) of rankings. (Best possible score = 1, Worst = 10).
> >
> > | Model                 | Claude 3.5 Sonnet Ranking (µ ± SD) | GPT-4o Ranking (µ ± SD) | Gemini 1.5 Pro Ranking (µ ± SD) | Combined Ranking (µ ± SD) |
> > |-----------------------|-------------------------|-----------------------|--------------------------|---------------------------|
> > | GPT-4 Turbo           | **2.27 ± 1.35**             | **2.13 ± 1.20**           | **2.07 ± 1.19**              | **2.16 ± 1.12**               |
> > | Gemini 1.5 Flash      | 3.09 ± 1.64             | 2.92 ± 1.63           | 2.98 ± 1.61              | 3.00 ± 1.28               |
> > | Claude 3 Sonnet       | 3.77 ± 2.61             | 4.14 ± 2.53           | 3.54 ± 2.31              | 3.82 ± 1.59               |
> > | Claude 3 Haiku        | 4.00 ± 2.11             | 4.37 ± 2.25           | 4.56 ± 1.98              | 4.31 ± 1.47               |
> > | Gemini 1.0 Pro Vision | 4.72 ± 2.16             | 4.22 ± 1.92           | 4.33 ± 1.87              | 4.42 ± 1.42               |
> > | True Caption          | 6.22 ± 2.03             | 5.72 ± 1.93           | 5.67 ± 2.17              | 5.87 ± 1.45               |
> > | Qwen VL Chat          | 5.86 ± 2.03             | 5.77 ± 1.67           | 6.06 ± 1.73              | 5.90 ± 1.35               |
> > | TransCore-M           | 7.85 ± 1.51             | 8.05 ± 1.37           | 8.09 ± 1.27              | 8.00 ± 1.18               |
> > | OmniLMM 3b            | 8.57 ± 1.27             | 8.75 ± 1.32           | 8.83 ± 1.13              | 8.72 ± 1.12               |
> > | Yi VL 6b              | 8.65 ± 1.53             | 8.93 ± 1.20           | 8.87 ± 1.22              | 8.82 ± 1.15               |
> >
> > 9. Observations:
> > - Clear delineation of preference among test models
> > - The closed-source models outperform the open-source models (as they do on the SciFIBench Figure->Caption task)
> > - Captions generated by all closed-source models are preferred over the True Caption. Conversely, the True Captions are preferred over all the open-source model captions.
> > - Strong agreement is shown between the evaluator models

---

> > > ### Comment · Reviewer_9Ksv · 2024-08-30
> > > **Maintain my score**
> > >
> > > Thank the authors for conducting the additional experiments and sharing the results. However,
> > > 1) Caption generation analysis: besides the statistics for model performance, it is also necessary to include a few specific examples of generated captions in the paper to illustrate the qualitative differences between models. This would help readers better understand what the numerical rankings mean in practice.
> > > 2) Dataset scale:  I would like to see more details on how you plan to integrate this new data while maintaining consistency and integrating it with the existing dataset.
> > > 3) Dataset statistics:  I would like to see more details on additional dataset statistics. This information will be crucial for users to understand the dataset's composition and potential biases.
> > > 4) Novelty in dataset construction: While the additional experiments and proposed expansions are valuable, the core methodology for dataset construction remains largely unchanged from previous work
> > >
> > >
> > > Given these considerations, I maintain my score

---

> > > > ### Author Rebuttal · Authors · 2024-08-31
> > > >
> > > > Thank you for your helpful comments, here are our responses:
> > > >
> > > > 1. Good suggestion – we have saved all of the generated captions and we will include examples in the final version of the paper.
> > > > 2. **Expanding SciFIBench scale and categories**
> > > >
> > > > Following the suggestions to increase both the scale and diversity of categories of SciFIBench, we have undertaken the following:
> > > > - Started with the recent ArxivCap dataset (570k papers, 6.4M images, covering these arXiv categories: cs, econ, eess, math, physics, q-bio, q-fin, stat)
> > > > - Randomly downselected 25%.
> > > > - Cleaned (removing fig-caption pairs with no caption text, removing CS papers, removing duplicates)
> > > > - Took category-balanced samples totalling ~102k figure-caption pairs
> > > > - Computed joint embeddings for each pair
> > > > - Found nearest neighbours for each figure-caption pair
> > > > - Constructed dataset of potential questions (102k)
> > > > - Sorted by difficulty (avg. distance)
> > > > - Hand-select a pool of answerable questions
> > > >
> > > > From this new pool we have curated a set of **844** high-quality, answerable questions. Here, we present accuracy results on this set of questions for the Figure->Caption and Caption->Figure tasks with adversarially selected negatives:
> > > >
> > > > | Model             | Accuracy (%)    |                 |   |
> > > > |-------------------|-----------------|-----------------|---|
> > > > |                   | Figure->Caption | Caption->Figure |   |
> > > > | GPT-4o            | 74.9            | 59.7            |   |
> > > > | GPT-4 Turbo       | 69.2            | 55.4            |   |
> > > > | Gemini 1.5 Pro    | 71.5            | 40.3            |   |
> > > > | Gemini 1.5 Flash  | 70.7            | 65.8            |   |
> > > > | Gemini Pro Vision | 53.3            | 37.7            |   |
> > > > | Claude 3 Sonnet   | 50.5            | 57.8            |   |
> > > > | Claude 3 Haiku    | 57.2            | 30.8            |   |
> > > >
> > > >
> > > > The results suggest questions in these new categories (econ, eess, math, physics, q-bio, q-fin, stat) are similarly difficult to those presented in the original paper, with the new Caption->Figure questions proving slightly more challenging, increasing the headroom of our benchmark. As we have now constructed the dataset of 102k questions covering these new categories, we just need to continue our rigorous human verification process to additionally increase the scale of questions that we add to SciFIBench. We plan to continue this to create a total of more than 2000 high-quality gold questions for SciFIBench.
> > > >
> > > > Moreover, in the final version we will include results on this expanded dataset for the full suite of models we previously evaluated.
> > > >
> > > > 3. For the final version of the paper, in addition to the statistics already provided, we promise to include a breakdown of (i) Figure type, (ii) Figure resolution, and (iii) Caption length.
> > > >
> > > >
> > > >
> > > > Hopefully, these experiments address the concerns you raised in your review regarding dataset scale and additional scenarios.

---

> > > > > ### Author Rebuttal · Authors · 2024-08-31
> > > > >
> > > > > Thank you again for your time reviewing our work. At the top of this page we have added a summary of our combined rebuttal to all the reviewers.

---

### Official Review · Reviewer_X2MV · 2024-08-01
**This paper provides Caption/Figure matching QA tasks from Arxiv CS papers for Multimodal Models evaluation.**

**Rating:** 5
**Confidence:** 5
**Correctness:** sound
**Clarity:** sound

**Review:**

quality: The provided 1000 gold questions are of high quality for testing the figure/caption matching abilities of current multimodal models. Human verification guarantee the quality of those 1000 questions.

clarity: this paper is well-written and easy-to-follow. The data is documented well and the whole data filtering pipeline is reasonable.

originality: The idea of using both caption->figure and figure->caption is novel and smart for a comprehensive test of multimodal models. Though the datasets only contain Arxiv CS papers, it turned our current models lag behind humans by a lot margin.

significance: Since the provided 1000 gold questions are valuable to prove the low performance of current multimodal models, this dataset is valuable for evaluating the current multimodal models. But only cs-related questions limited its usefulness. Also, there is no evidence the large-scale noisy data is really useful.

**Strengths:**

See reviews above.

**Additional Feedback:**

1. Any reason for why Caption → Figure is more difficult than Figure → Caption.?

**Documentation:**

yes

**Limitations:**

There is no evidence that the Bronze and Silver data would improve the models if additional continual training are performed on them. Can you train the models on those large-scale models to prove their usefulness? The reason is that those data is very noisy, so I am unsure whether they are valuable.

Can you also include more broad categories of arXiv papers? The current data only contain CS-related questions and are thus limited.

**Opportunities For Improvement:**

1. lack of error analysis. It would be great to provide a detailed analysis of which categories of questions are more difficult, leading to more errors? Also, how does each model make errors? and what's their difference in error pattern? Since the dataset contains arxiv paper from 2010-2020, does the error correlates with time? What's the difference between human's error pattern and models?

2. There is no evidence that the Bronze and Silver data would improve the models if additional continual training are performed on them. Can you train the models on those large-scale models to prove their usefulness? The reason is that those data is very noisy, so I am unsure whether they are valuable.

3. Can you also include more broad categories of arXiv papers? The current data only contain CS-related questions and are thus limited.

**Relation To Prior Work:**

yes

**Summary And Contributions:**

This paper provides SciFIBench, a scientific figure interpretation benchmark consisting of a gold set of 1000 CS-related questions split between two tasks across 12 categories. They evaluate 27 LMMs on SciFIBench, finding it challenging. However, humans can achieve near-perfect performance on those, showing that current models still have a large space for improvements. They also provide additional large-scale noisy data, which is potentially useful for pertaining.

---

> ### Author Rebuttal · Authors · 2024-08-16
>
> Thank you for your insightful comments that will help strengthen our paper.  We are pleased that you found our submission well-written and easy-to-follow and that SciFIBench is valuable for evaluating the current multimodal models. In the following, we include responses to the concerns raised in your review.
>
> **Lack of error analysis**: In the Supplementary material, we included a per-category performance comparison for a selection of evaluated models. We can incorporate this analysis (Supp. Fig. 2) into the main paper. Thank you for your other error analysis suggestions. We will update the main paper to include: (1) a qualitative analysis of the types of error each model and humans make, (2) analysis of error rate vs. publication date, (3) analysis of error rate vs. figure resolution.
>
> **Improved performance by using bronze + silver datasets**: Thank you for raising the important point about the potential benefits of using the noisier bronze and silver data sets for fine-tuning or few-shot learning. We have conducted additional experiments using Gemini-Pro Vision in a few-shot setting with questions from the silver set to explore this issue further.
>
> In our experiments, we tested the model in a 0-shot setting using the same prompt outlined in our paper, and in k-shot settings where we appended the following to the original prompt:
>
> “””
> Your task is to identify the correct caption for the figure. Here are some potentially noisy examples to help you:
>
> [Following repeated k times]
> <EXAMPLE>
> {Example Fig}{Captions}{Question}{Correct Answer}
> </EXAMPLE>
> “””
>
> Here are the results from two experimental settings (1) ‘Similar examples’ in which k-shot examples were sampled from the Silver set based on similarity to the question and (2) ‘Random examples’ in which k-shot samples were randomly selected from the Silver set. The random examples experiments were repeated 5 times with different random seeds (standard deviation reported in the table).
>
> | k (num examples)  | Similar examples | Random examples (mean from 5 repeats) | Random examples (s.d. from 5 repeats) |
> |----|------------------|---------------------------------------|---------------------------------------|
> | 0  | 56.8%            | 56.8%                                 | N/A                                  |
> | 1  | 57.6%            | 55.8%                                 | 1.2%                                  |
> | 2  | 61.2%            | 59.4%                                 | 0.6%                                  |
> | 3  | 59.2%            | 58.4%                                 | 1.0%                                  |
> | 4  | 58.0%            | 58.3%                                 | 1.0%                                  |
> | 5  | 58.0%            | 58.0%                                 | 1.4%                                  |
>
>
> We found that model performance is sensitive to the prompt. With some prompt structures, the presence of examples decreases performance. However, in the case outlined above, we observe that incorporating examples from the silver set can quantitatively improve performance compared to the 0-shot setting. Additionally, a qualitative review of the outputs suggests that the model's responses more closely follow the instructed format when examples are included, which reduces the need for an LLM to parse the correct answer from the LMM output.
>
> We acknowledge that the silver/bronze set noisiness can impact performance, particularly when examples are not well-matched to the question. Nonetheless, these results indicate that, with the right prompt structure, the silver data can be valuable in improving model performance. We will update our manuscript to include these findings and to discuss the potential trade-offs when using noisy data. Furthermore, our results align with observations from the MMMU benchmark (https://arxiv.org/pdf/2311.16502), which also noted performance can decline with increased shots in certain scenarios.
>
> **Inclusion of broader categories**: Thank you for raising this. We agree that it would be beneficial to include more categories as part of SciFIBench. To address this, we propose to expand SciFIBench by leveraging the ArxivCap data (https://huggingface.co/datasets/MMInstruction/ArxivCap, which includes 32 scientific domains), focusing on the arXiv categories not included in SciFIBench. We will construct the questions using the methodology outlined in the SciFIBench paper. We aim to provide preliminary results for questions from these new categories before the end of the reviewer/author discussion period.
>
> **Additional feedback**: Multi-image tasks (Caption->Figure) are known to be challenging to LMMs and typically more difficult than the singe-image setting (Figure->Caption). Enhancing a model’s multi-image ability can cause performance degeneration in the single-image setting (e.g., https://arxiv.org/abs/2405.01483). We will add an outline of these findings to our final paper.

---

> > ### Author Rebuttal · Authors · 2024-08-31
> >
> > **Expanding SciFIBench scale and categories**
> >
> > Following the suggestions to increase both the scale and diversity of categories of SciFIBench, we have undertaken the following:
> > - Started with the recent ArxivCap dataset (570k papers, 6.4M images, covering these arXiv categories: cs, econ, eess, math, physics, q-bio, q-fin, stat)
> > - Randomly downselected 25%.
> > - Cleaned (removing fig-caption pairs with no caption text, removing CS papers, removing duplicates)
> > - Took category-balanced samples totalling ~102k figure-caption pairs
> > - Computed joint embeddings for each pair
> > - Found nearest neighbours for each figure-caption pair
> > - Constructed dataset of potential questions (102k)
> > - Sorted by difficulty (avg. distance)
> > - Hand-select a pool of answerable questions
> >
> > From this new pool we have curated a set of **844** high-quality, answerable questions. Here, we present accuracy results on this set of questions for the Figure->Caption and Caption->Figure tasks with adversarially selected negatives:
> >
> > | Model             | Accuracy (%)    |                 |   |
> > |-------------------|-----------------|-----------------|---|
> > |                   | Figure->Caption | Caption->Figure |   |
> > | GPT-4o            | 74.9            | 59.7            |   |
> > | GPT-4 Turbo       | 69.2            | 55.4            |   |
> > | Gemini 1.5 Pro    | 71.5            | 40.3            |   |
> > | Gemini 1.5 Flash  | 70.7            | 65.8            |   |
> > | Gemini Pro Vision | 53.3            | 37.7            |   |
> > | Claude 3 Sonnet   | 50.5            | 57.8            |   |
> > | Claude 3 Haiku    | 57.2            | 30.8            |   |
> >
> >
> > The results suggest questions in these new categories (econ, eess, math, physics, q-bio, q-fin, stat) are similarly difficult to those presented in the original paper, with the new Caption->Figure questions proving slightly more challenging, increasing the headroom of our benchmark. As we have now constructed the dataset of 102k questions covering these new categories, we just need to continue our rigorous human verification process to additionally increase the scale of questions that we add to SciFIBench. We plan to continue this to create a total of more than 2000 high-quality gold questions for SciFIBench.
> >
> > Moreover, in the final version we will include results on this expanded dataset for the full suite of models we previously evaluated.

---

> > > ### Author Rebuttal · Authors · 2024-08-31
> > >
> > > Thank you again for your time reviewing our work. At the top of this page we have added a summary of our combined rebuttal to all the reviewers.

---

### Comment · Area_Chair_d5qz · 2024-08-29
**Reminder to review comments before end of discussion period on 8/31**

Reviewers, thank you for your time and contributions thus far. This is a reminder that the discussion period ends in two days on August 31. Please take some time to engage with the authors' comments and adjust scores if appropriate.

---

> ### Author Rebuttal · Authors · 2024-08-30
>
> Dear Reviewers,
>
> We are currently finishing off experimentation on an expanded question set of new arXiv categories and will share the results shortly.

---

### Author Rebuttal · Authors · 2024-08-31

Dear Reviewers,

Thank you for your efforts reviewing our paper and your positive comments. We appreciate your suggestions for improvements and opportunities to strengthen our paper. During this rebuttal period we have invested a lot of effort to address your concerns. As different concerns were raised by each of you, we will summarise our rebuttal here:

*What we have done*:

**Scale and diversity**:  We have increased the scale of SciFIBench by adding 844 new gold-questions covering econ, eess, math, physics, q-bio, q-fin, stat arXiv categories. These 844 were hand-selected from a pool of 102k, from which we will continue to add more. On the Figure-Caption task, we have found this new question set to be of similar difficulty to the original CS SciFIBench questions, while on the Caption-Figure task they prove more challenging.
(See response to Reviewers X2MV and 9Ksv for details).

**Benefits of the bronze and silver sets**: We conducted few-shot experiments using examples from the silver set and found that incorporating these examples does quantitatively improve performance on the gold set questions. Furthemore, the examples also more closely align the model output with the format instructions, reducing the need for LLM evaluation. (See response to Reviewer X2MV for details).

**Captions generation**: On a subset of SciFIBench, we carried out a study comparing the quality of captions generated by a set of LMMs as evaluated using a holdout set of strong LMMs. We found a clear delineation of preference among the test models, with the closed-source models outperforming the open-source models and providing captions preferred over the ground truth caption. Strong agreement was shown between the evaluator models.
(See response to Reviewer 9Ksv for more details).

*What we will do*:

**Error analysis**: Across all our experimentation we have saved the model output. We will use this to conduct error analysis: (1) Across categories, (2) Types of error, (3) Error rate vs. publication date, (4) Error rate vs. Figure resolution, (5) Error rate vs. Caption length. This will be included in the final paper.

**Analytics/statistics**: In addition to the provided statistics, we will also include a breakdown of (i) Figure type, (ii) Figure resolution, and (iii) Caption length.

We sincerely appreciate the reviewers' feedback and the opportunity to clarify and improve our work. We hope that the additional experiments we have conducted during this rebuttal address your main concerns and also provide confidence that we will deliver on our remaining promises for the final version of the paper. We kindly ask that you consider these improvements when reevaluating our work.

---

### Decision · Program_Chairs · 2024-09-26

**Decision:**

Accept (Poster)

**Comment:**

**Quality:** The paper introduces a well-executed benchmark for evaluating LLMs for interpreting scientific figure. The dataset contains a substantially-sized, human curated set of figure-caption pairs and multiple choice questions. Experiments are comprehensive, testing 27 LLMs under a variety of conditions. Data and code are shared publicly.

**Clarity:** The paper is clearly written.

**Originality:** While the dataset construction and figure-caption matching tasks are not particularly novel, the quality, size, and utility of the benchmark are. The benchmark will allow for better evaluation of multimodal LLMs for scientific figure tasks.

**Significance:** The authors have conducted comprehensive experiments across a range of models and conditions to demonstrate the utility of this benchmark, adding value that has been acknowledged by all reviewers.

I commend the authors on expanding the scope of the dataset and conducting additional experiments to address reviewer feedback. I believe all reviewer comments have been addressed or could be addressed in a camera ready revision.